# Kernel-Based Tests for Likelihood-Free Hypothesis Testing

**Patrik Róbert Gerber**[*]
Department of Mathematics, MIT
Cambridge, MA 02139
prgerber@mit.edu

**Tianze Jiang**[*]
Department of Mathematics, MIT
Cambridge, MA 02139
tjiang@mit.edu

**Yury Polyanskiy**[*]
Department of EECS, MIT
Cambridge, MA 02139
yp@mit.edu

**Rui Sun**[*]
Department of Mathematics, MIT
Cambridge, MA 02139
eruisun@mit.edu

## Abstract

Given $n$ observations from two balanced classes, consider the task of labeling an additional $m$ inputs that are known to all belong to *one* of the two classes. Special cases of this problem are well-known: with complete knowledge of class distributions ($n = \infty$) the problem is solved optimally by the likelihood-ratio test; when $m = 1$ it corresponds to binary classification; and when $m \approx n$ it is equivalent to two-sample testing. The intermediate settings occur in the field of likelihood-free inference, where labeled samples are obtained by running forward simulations and the unlabeled sample is collected experimentally. In recent work it was discovered that there is a fundamental trade-off between $m$ and $n$: increasing the data sample $m$ reduces the amount $n$ of training/simulation data needed. In this work we (a) introduce a generalization where unlabeled samples come from a mixture of the two classes – a case often encountered in practice; (b) study the minimax sample complexity for non-parametric classes of densities under *maximum mean discrepancy* (MMD) separation; and (c) investigate the empirical performance of kernels parameterized by neural networks on two tasks: detection of the Higgs boson and detection of planted DDPM generated images amidst CIFAR-10 images. For both problems we confirm the existence of the theoretically predicted asymmetric $m$ vs $n$ trade-off.

## 1 Likelihood-Free Inference

The goal of likelihood-free inference (LFI) [8; 12; 17; 24], also called simulation-based inference (SBI), is to perform statistical inference in a setting where the data generating process is a black-box, but can be simulated. Given the ability to generate samples $X_\theta \sim P_\theta^{\otimes n}$ for any parameter $\theta$, and given real-world data $Z \sim P_{\theta^\star}^{\otimes m}$, we want to use our simulations to learn about the truth $\theta^\star$. LFI is particularly relevant in areas of science where we have precise but complex laws of nature, for which we can do (stochastic) forward simulations, but can not directly compute the (distribution) density $P_\theta$. The Bayesian community approached the problem under the name of Approximate Bayesian Computation (ABC) [6; 13; 47]. More recent ML-based methods where regressors and classifiers

---

[*]Equal contribution.

37th Conference on Neural Information Processing Systems (NeurIPS 2023).

are used to summarize data, select regions of interest, approximate likelihoods or likelihood-ratios [14; 29; 30; 42; 49] have also emerged for this challenge.

Despite empirical advances, the theoretical study of frequentist LFI is still in its infancy. We focus on the nonparametric and non-asymptotic setting, which we justify as follows. For applications where tight error control is critical one might be reluctant to rely on asymptotics. More broadly, the non-asymptotic regime can uncover new phenomena and provide insights for algorithm design. Further, parametric models are clearly at odds with the black-box assumption. Recently, [18] proposed likelihood-free hypothesis testing (LFHT) as a simplified model and found minimax optimal tests for a range of nonparametric distribution classes, thereby identifying a *fundamental simulation-experimentation trade-off* between the number of simulated observations $n$ and the size of the experimental data sample $m$. Here we extend [18], and prior related work [19; 26; 27; 32; 33], to a new setting designed to model experimental setups more truthfully and derive sample complexity (upper and lower bounds) for kernel-based tests over nonparametric classes.

While minimax optimal, the algorithms of [18; 19] are impractical as they rely on discretizing the observations on a regular grid. Thus, both in our theory as well as experiments we turn to kernel methods which provide an empirically more powerful set of algorithms that have shown success in nonparametric testing [20; 21; 22; 31; 39].

**Contributions** Our contributions are twofold. *Theoretically*, we introduce *mixed likelihood-free hypothesis testing* (mLFHT), which is a generalization of (LFHT) and provides a better model of applications such as the search for new physics [11; 38]. We propose a robust kernel-based test and derive both upper and lower bounds on its minimax sample complexity over a large nonparametric class of densities, generalizing multiple results in [18; 19; 26; 27; 32; 33; 37]. Although the simulation-experimentation ($m$ vs $n$) trade-off has been proven in the minimax sense (that is, for some worst-case data distribution), it is not clear whether it actually occurs in real data. Our second contribution is the *empirical* confirmation of the existence of an asymmetric trade-off, cf. Figure 1.2. To this end we construct state-of-the-art tests building on ideas of [39; 48] on learning good kernels from the data. We execute this program in two settings: the Higgs boson discovery [5], and detecting diffusion [25] generated images planted in the CIFAR-10 [35] dataset.

## 1.1 LFHT **and the Simulation-Experimentation Trade-off**

Suppose we have i.i.d. samples $X, Y$ each of size $n$ from two unknown distributions $P_X, P_Y$ on a measurable space $\mathcal{X}$, as well as a third i.i.d. sample $Z \sim P_Z$ of size $m$. In the context of LFI, we may think of the samples $X, Y$ as being generated by our simulator, and $Z$ being the data collected in the real world. The problem we refer to as likelihood-free hypothesis testing is the task of deciding between the hypotheses

$$H_0 : P_Z = P_X \quad \text{versus} \quad H_1 : P_Z = P_Y. \quad \text{(LFHT)}$$

This problem originates in [23; 52], where authors study the exponents of error decay for finite $\mathcal{X}$ and fixed $P_X, P_Y$ as $n \sim m \to \infty$; more recently [2; 18; 19; 26; 27; 32; 33; 37] it is studied in the non-asymptotic regime. Assuming that $P_X, P_Y$ belong to a known nonparametric class of distributions $\mathcal{P}$ and are guaranteed to be $\epsilon$-separated with respect to total variation (TV) distance (i.e. $\text{TV}(P_X, P_Y) \geq \epsilon$), [18] characterizes the sample sizes $n$ and $m$ required for the sum of type-I and type-II errors to be small, as a function of $\epsilon$ and for several different $\mathcal{P}$'s. Their results show, for three settings of $\mathcal{P}$, that $(i)$ testing (LFHT) at vanishing error is possible even when $n$ is not large enough to estimate $P_X$ and $P_Y$ within total variation distance $\mathcal{O}(\epsilon)$, and that $(ii)$ to achieve a fixed level of error, say $\alpha$, one can *trade off* $m$ vs. $n$ along a curve of the form $\{\min\{n, \sqrt{mn}\} \gtrsim n_{\text{TS}}(\alpha, \epsilon, \mathcal{P}), m \gtrsim \log(1/\alpha)/\epsilon^2\}$. Here $n_{\text{TS}}$ denotes the minimax sample complexity of two-sample testing over $\mathcal{P}$, i.e. the minimum number of observations $n$ needed from $P_X, P_Y \in \mathcal{P}$ to distinguish the cases $\text{TV}(P_X, P_Y) \geq \epsilon$ versus $P_X = P_Y$. Here $\gtrsim$ suppresses dependence on constants and untracked parameters.

It is unclear, however, whether predictions drawn from minimax sample complexities over specified distribution classes can be observed in real-world data. Without the theory, a natural expectation is that the error contour $\{(m, n) : \exists \text{ a test with total error} \leq \alpha\}$ would look similar to that of minimax two-sample testing with unequal sample size, namely $\{(m, n) : \min\{n, m\} \gtrsim n_{\text{TS}}(\alpha, \epsilon, \mathcal{P})\}$, i.e. $n$ and $m$ simply need to be above a certain threshold simultaneously (as is the case for e.g. two-sample testing over smooth densities [4; 37]). However, from Figures 2 and 1.2 we see that there is indeed a

non-trivial trade-off between $n$ and $m$: the contours are not always parallel to the axes and aren't symmetric about the line $m = n$. The importance of Fig. 1.2 is in demonstrating that said trade-off is not a kink of a theory that arises due to some esoteric worst-case data distribution, but is instead a real effect observed in state-of-the-art LFI algorithms ran on actual data. We remark that the $n$ used in this plot is the total number of simulated samples (most of which are used for choosing a neural-network parameterized kernel) and are not just the $n$ occuring in Theorems 3.1 and 3.2 which apply to a *fixed* kernel. See Section 4 for details on sample division.

## 1.2 Mixed Likelihood-Free Hypothesis Testing

A prominent application of likelihood-free inference lies in the field of particle physics. Scientists run sophisticated experiments in the hope of finding a new particle or phenomenon. Often said phenomenon can be predicted from theory, and thus can be simulated, as was the case for the Higgs boson whose existence was verified after nearly 50 years at the Large Hadron Collider (LHC) [3; 9].

Suppose we have $n$ simulations from the *background* distribution $P_X$ and the *signal* distribution $P_Y$. Further, we also have $m$ (real-world) datapoints from $P_Z = (1 - \nu)P_X + \nu P_Y$, i.e. the observed data is a mixture between the background and signal distributions with rate parameter $\nu$. The goal of physicists is to construct confidence intervals for $\nu$, and a *discovery* corresponds to a $5\sigma$ confidence interval that excludes $\nu = 0$. We model this problem by testing

$$H_0 : \nu = 0 \quad \text{versus} \quad H_1 : \nu \geq \delta \qquad \text{(mLFHT)}$$

for fixed (usually predicted) $\delta > 0$. See the rigorous definition of (mLFHT) in Section 3. In particular, a discovery can be claimed if $H_0$ is rejected.

## 2 The Likelihood-Free Test Statistic

This section introduces the testing procedure based on Maximum Mean Discrepancy (MMD) that we study throughout the paper both theoretically and empirically. First, we introduce the necessary background on MMD in Section 2.1. Then, we define our test statistics in Section 2.2.

## 2.1 Kernel Embeddings and MMD

Given a set $\mathcal{X}$, we call the function $K : \mathcal{X}^2 \to \mathbb{R}$ a kernel if the $n \times n$ matrix with $ij$'th entry $K(x_i, x_j)$ is symmetric positive semidefinite for all choices of $x_1, \ldots, x_n \in \mathcal{X}$ and $n \geq 1$. There is a unique reproducing kernel Hilbert space (RKHS) $\mathcal{H}_K$ associated to $K$. $\mathcal{H}_K$ consists of functions $\mathcal{X} \mapsto \mathbb{R}$ and satisfies the reproducing property $\langle K(x, \cdot), f \rangle_{\mathcal{H}_K} = f(x)$ for all $f \in \mathcal{H}_k$ and $x \in \mathcal{X}$, in particular $K(x, \cdot) \in \mathcal{H}_K$. Given a probability measure $P$ on $\mathcal{X}$, define its kernel embedding $\theta_P$ as

$$\theta_P \triangleq \mathbb{E}_{X \sim P} K(X, \cdot) = \int_{\mathcal{X}} K(x, \cdot)P(\mathrm{d}x). \qquad (1)$$

Given the kernel embeddings of two probability measures $P, Q$, we can measure their distance in the RKHS by $\mathrm{MMD}(P, Q) \triangleq \|\theta_P - \theta_Q\|_{\mathcal{H}_K}$, where MMD stands for maximum mean discrepancy. MMD has a closed form thanks to the reproducing property and linearity:

$$\mathrm{MMD}^2(P, Q) = \mathbb{E}\left[K(X, X') + K(Y, Y') - 2K(X, Y)\right]$$

where $(X, X', Y, Y') \sim P^{\otimes 2} \otimes Q^{\otimes 2}$. In particular, if $P, Q$ are empirical measures based on observations, we can evaluate the MMD exactly, which is crucial

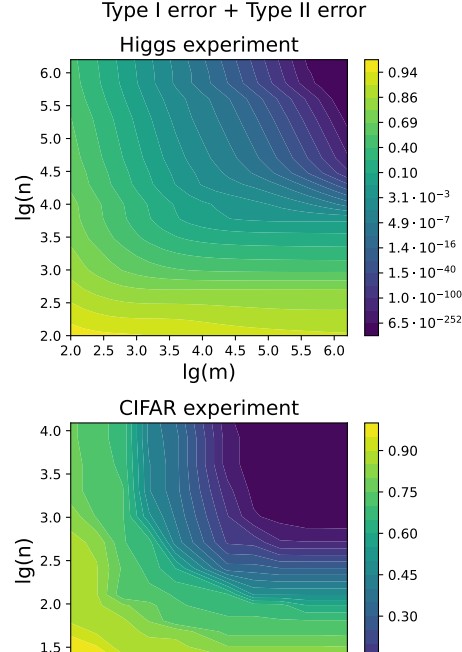

Type I error + Type II error

Figure 1: $n$ versus $m$ trade-off for the Higgs and CIFAR experiments using our test in Section 2. Error probabilities are estimated by normal approximation for Higgs and simulated for CIFAR.

in practice. Yet another attractive property of MMD is that (under mild integrability conditions) it is an integral probability metric (IPM) where the supremum is over the unit ball of the RKHS $\mathcal{H}_K$. See e.g. [41; 46] for references. The following result is a consequence of the fact that self-adjoint compact operators are diagonalizable.

**Theorem 2.1** (Hilbert–Schmidt). *Suppose that $K \in L^2(\mu \otimes \mu)$ is symmetric. Then there exists a sequence $(\lambda_j)_{j \geq 1} \in \ell^2$ and an orthonormal basis $\{e_j\}_{j \geq 1}$ of $L^2(\mu)$ such that $K(x,y) = \sum_{j \geq 1} \lambda_j e_j(x) e_j(y)$ for all $j \geq 1$, where convergence is in $L^2(\mu \otimes \mu)$.*

**Assumption 1.** Unless specified otherwise, we implicitly assume a choice of a non-negative measure $\mu$ and kernel $K$ for which the conditions of Theorem 2.1 hold. Note that $\lambda_j \geq 0$ and depend on $\mu$.

**Removing the Bias**  In our proofs we work with the kernel embedding of empirical measures for which we need to modify the inner product $\langle \cdot, \cdot \rangle_{\mathcal{H}_K}$ (and thus MMD) slightly by removing the diagonal terms. Namely, given i.i.d. samples $X, Y$ of size $n, m$ respectively and corresponding empirical measures $\widehat{P}_X, \widehat{P}_Y$, we define

$$\mathrm{MMD}_u^2(\widehat{P}_X, \widehat{P}_Y) \triangleq \sum_{i \neq j} \frac{K(X_i, X_j)}{n(n-1)} + \sum_{i \neq j} \frac{K(Y_i, Y_j)}{m(m-1)} - 2 \sum_{i,j} \frac{K(X_i, Y_j)}{mn}. \tag{2}$$

We also write $\langle \theta_{\widehat{P}_X}, \theta_{\widehat{P}_X} \rangle_{u, \mathcal{H}_K} \triangleq \|\theta_{\widehat{P}_X}\|_{u, \mathcal{H}_K}^2 \triangleq \frac{1}{n(n-1)} \sum_{i \neq j} K(X_i, X_j)$ and extend linearly. The $u$ stands for unbiased, since $\mathbb{E}\,\mathrm{MMD}_u^2(\widehat{P}_X, \widehat{P}_Y) = \mathrm{MMD}^2(P_X, P_Y) \neq \mathbb{E}\,\mathrm{MMD}^2(\widehat{P}_X, \widehat{P}_Y)$ in general.

## 2.2  Test Statistic

With Section 2.1 behind us, we are in a position to define the test statistic that we use to tackle (mLFHT). Suppose that we have samples $X, Y, Z$ of sizes $n, n, m$ from the probability measures $P_X, P_Y, P_Z$. Write $\widehat{P}_X$ for the empirical measure of sample $X$, and analogously for $Y, Z$. The core of our test statistic for (mLFHT) is the following:

$$T(X, Y, Z) \triangleq \langle \theta_{\widehat{P}_Z}, \theta_{\widehat{P}_Y} - \theta_{\widehat{P}_X} \rangle_{u, \mathcal{H}_K} = \frac{1}{nm} \sum_{i=1}^{n} \sum_{j=1}^{m} \left\{ K(Z_j, Y_i) - K(Z_j, X_i) \right\}. \tag{3}$$

Note that $T$ is of the additive form $\frac{1}{m} \sum_{j=1}^{m} f(Z_i)$ where $f(z) \triangleq \theta_{\widehat{P}_Y}(z) - \theta_{\widehat{P}_X}(z)$ can be interpreted as the *witness function* of [21; 31]. Given some $\pi \in [0, 1]$ (taken to be half the predicted signal rate $\delta/2$ in our proofs), the output of our test is

$$\Psi_\pi = \mathbb{1}\left\{ T(X, Y, Z) \geq \gamma(X, Y, \pi) \right\}, \text{ where } \gamma(X, Y, \pi) = \pi\,\mathrm{MMD}_u^2(\widehat{P}_X, \widehat{P}_Y) + T(X, Y, X). \tag{4}$$

The threshold $\gamma$ gives $\Psi_\pi$ a natural geometric interpretation: it checks whether the projection of $\theta_{\widehat{P}_Z} - \theta_{\widehat{P}_X}$ onto the vector $\theta_{\widehat{P}_Y} - \theta_{\widehat{P}_X}$ falls further than $\pi$ along the segment joining $\theta_{\widehat{P}_X}$ to $\theta_{\widehat{P}_Y}$ (up to deviations due to the omitted diagonal terms, see Section 2.1).

Setting $\delta = 1$ in (mLFHT) recovers (LFHT), and the corresponding test output is $\Psi_{\delta/2} = \Psi_{1/2} = 1$ if and only if $\mathrm{MMD}_u(\widehat{P}_Z, \widehat{P}_X) \geq \mathrm{MMD}_u(\widehat{P}_Z, \widehat{P}_Y)$. This very statistic (i.e. $\mathrm{MMD}_u(\widehat{P}_Z, \widehat{P}_X) - \mathrm{MMD}_u(\widehat{P}_X, \widehat{P}_Y)$) has been considered in the past for relative goodness-of-fit testing [7] where it's asymptotic properties are established. In the non-asymptotic setting, if MMD is replaced by the $L^2$-distance we recover the test statistic studied by [18; 27; 33]. However, we are the first to introduce $\Psi_\delta$ for $\delta \neq 1$ and to study MMD-based tests for (m)LFHT in a non-asymptotic setting.

**Variance cancellation**  At first sight it may seem more natural to the reader to threshold the distance $\mathrm{MMD}_u(\widehat{P}_Z, \widehat{P}_X)$, resulting in rejection if, say, $\mathrm{MMD}_u(\widehat{P}_Z, \widehat{P}_X) \geq \mathrm{MMD}_u(\widehat{P}_X, \widehat{P}_Y)\delta/2$. The geometric meaning of this would be similar to the one outlined above. However, there is a crucial difference: (LFHT) (the case $\delta = 1$) is possible with very little experimental data $m$ due to the *cancellation of variance*. More precisely, the statistic $\mathrm{MMD}^2(\widehat{P}_Z, \widehat{P}_X)$ contains the term $\frac{1}{m(m-1)} \sum_{i \neq j} K(Z_i, Z_j)$ — whose variance is prohibitively large and would inflate the $m$ required for reliable testing — but this can be canceled by subtracting $\mathrm{MMD}^2(\widehat{P}_Z, \widehat{P}_Y)$. Our statistic $T(X, Y, Z) - \gamma(X, Y, \pi)$ simply generalizes this idea to (mLFHT).

# 3 Minimax Rates of Testing

## 3.1 Upper Bounds on the Minimax Sample Complexity of (mLFHT)

Let us start by reintroducing (mLFHT) in a rigorous fashion. Given $C, \epsilon, R \geq 0$, let $\mathcal{P}_\mu(C, \epsilon, R)$ denote the set of triples $(P_X, P_Y, P_Z)$ of distributions such that the following three conditions hold:

(i) $P_X, P_Y$ and $P_Z$ have $\mu$-densities bounded by $C$,

(ii) $\mathrm{MMD}(P_X, P_Y) \geq \epsilon$, and

(iii) $\mathrm{MMD}(P_Z, (1-\nu)P_X + \nu P_Y) \leq R \cdot \mathrm{MMD}(P_X, P_Y)$,

where we define $\nu = \nu(P_X, P_Y, P_Z) = \arg\min_{\nu' \in \mathbb{R}} \mathrm{MMD}(P_Z, (1-\nu')P_X + \nu' P_Y)$. For some $\delta > 0$, consider the two hypotheses

$$
\begin{aligned}
H_0(C, \epsilon, \delta, R) &: (P_X, P_Y, P_Z) \in \mathcal{P}_\mu(C, \epsilon, R) \text{ and } \nu = 0 \\
H_1(C, \epsilon, \delta, R) &: (P_X, P_Y, P_Z) \in \mathcal{P}_\mu(C, \epsilon, R) \text{ and } \nu \geq \delta,
\end{aligned}
\tag{5}
$$

which we regard as subsets of probability measures. Notice that $R$ controls the level of mis-specification in the direction that is orthogonal to the line connecting the kernel embeddings of $P_X$ and $P_Y$. Setting $R = 0$ simply asserts that $P_Z$ is guaranteed to be a mixture of $P_X$ and $P_Y$, as is the case for prior works on LFHT. Before presenting our main result on the minimax sample complexity of mLFHT, let us define one final piece of terminology. We say that a test $\Psi$, which takes some data as input and takes values in $\{0, 1\}$, has total error probability less than $\alpha$ for the problem of testing $H_0$ vs $H_1$ if

$$
\sup_{P \in H_0} P(\Psi = 1) + \sup_{Q \in H_1} Q(\Psi = 0) \leq \alpha.
\tag{6}
$$

**Theorem 3.1.** *Suppose we observe three i.i.d. samples $X, Y, Z$ from distributions $P_X, P_Y, P_Z$ composed of $n, n, m$ observations respectively and let $C \in (0, \infty)$ and $R, \epsilon \geq 0$ and $\delta \in (0, 1)$. There exists a universal constant $c > 0$ such that $\Psi_{\delta/2}$ defined Section 2.2 tests $H_0$ vs $H_1$, as defined in (5), at total error $\alpha$ provided*

$$
\min\{m, n\} \geq c \frac{C\|\lambda\|_\infty \log(1/\alpha)}{(\epsilon\delta/(1+R))^2} \qquad and \qquad \min\{n, \sqrt{nm}\} \geq c \frac{C\|\lambda\|_2 \log(1/\alpha)}{\delta\epsilon^2}.
$$

Note that Theorem 3.1 does *not* place assumptions on the distributions $P_X, P_Y$ beyond bounded density with respect to the base measure $\mu$. This is different from usual results in statistics, where prior specification of distribution classes is crucial. On the other hand, instead of standard distances such as $L^p$, we assume separation with respect to MMD and the latter is potentially harder to interpret than, say, $L^1$ i.e. total variation. We do point out that our Theorem 3.1 can be used to derive results in the classical setting; we discuss this further in Section 3.4.

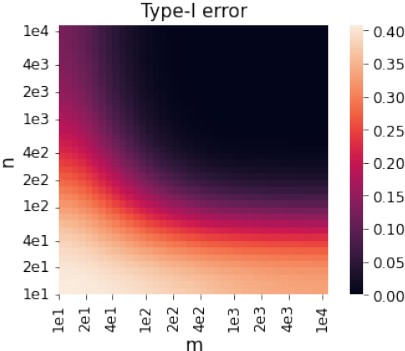

Figure 2: $n$ versus $m$ trade-off for the toy experiment, verifying Theorem 3.1. Probabilities estimated over $10^4$ runs, and smoothed using Gaussian noise.

In an appropriate regime of the parameters, the sufficient sample complexity in Theorem 3.1 exhibits a trade-off of the form $\min\{n, \sqrt{mn}\} \gtrsim \|\lambda\|_2 \log(1/\alpha)/(\delta\epsilon^2)$ between the number of simulation samples $n$ and real observations $m$. This trade-off is shown in Figure 2 using data from a toy problem. The trade-off is clearly asymmetric and the relationship $m \cdot n \geq const.$ also seems to appear. In this toy problem we set $R = 0, \delta = 1, \epsilon = .3, k = 100$ and $P_X = P_Z, P_Y$ are distributions on $\{1, 2, \ldots, k\}$ with $P_X(i) = (1 + \epsilon \cdot (2 \cdot \mathbb{1}\{i \text{ odd}\} - 1))/k = 2/k - P_Y(i)$ for all $i = 1, 2, \ldots, k$. The kernel we take is $K(x, y) = \sum_{i=1}^{k} \mathbb{1}\{x = y = i\}$ and $\mu$ is simply the counting measure; the resulting MMD is simply the $L^2$-distance on pmfs.

Figure 1.2 illustrates a larger scale experiment using real data using a trained kernel. Note that we plot the *total* number $n$ of simulation samples, including those used for *training* the kernel itself (see Section 4); which ensures that Figure 1.2 gives a realistic picture of data requirements. However, due to the dependence between the kernel and the data, Theorem 3.1 no longer applies. Nevertheless, we observe a trade-off similar to Figure 2.

## 3.2 Lower Bounds on the Minimax Sample Complexity of (mLFHT)

In this section we prove a minimax lower bound on the sample complexity of mLFHT, giving a partial converse to Theorem 3.1. Before we can state this results, we must make some technical definitions. Given $J \geq 2$, let $\|\lambda\|_{2J}^2 \triangleq \sum_{j=2}^{J} \lambda_j^2$ and define

$$J_\epsilon^\star \triangleq \max \left\{ J : \sup_{\eta_j = \pm 1} \Big\| \sum_{j=2}^{J} \eta_j \sqrt{\lambda_j} e_j \Big\|_\infty \leq \frac{\|\lambda\|_{2J}}{2\epsilon} \right\}.$$

**Theorem 3.2** (Lower Bounds for mLFHT). *Suppose that $\int_{\mathcal{X}} K(x,y)\mu(\mathrm{d}x) \equiv \lambda_1$, $\mu(\mathcal{X}) = 1$ and $\sup_{x \in \mathcal{X}} K(x,x) \leq 1$. There exists a universal constant $c > 0$ such that any test of $H_0$ vs $H_1$, as defined in (5), with total error at most $\alpha$ must use a number $(n,m)$ of observations that satisfy*

$$m \geq c \frac{\lambda_2 \log(1/\alpha)}{\epsilon^2 \delta^2} \quad and \quad n \geq c \frac{\|\lambda\|_{2J_\epsilon^\star} \sqrt{\log(1/\alpha)}}{\epsilon^2} \quad and \quad \delta m + \sqrt{mn} \geq c \frac{\|\lambda\|_{2J_\epsilon^\star} \sqrt{\log(1/\alpha)}}{\epsilon^2 \delta}.$$

*Remark* 3.3. Recall that the eigenvalues $\lambda$ *depend on the choice of* $\mu$, so that by choosing a different base measure $\mu$ one can optimize the lower bound. However, since $P_X, P_Y, P_Z$ are assumed to have bounded density with respect to $\mu$, this appears rather involved.

*Remark* 3.4. The requirements $\sup_{x \in \mathcal{X}} K(x,x) \leq 1$ and $\mu(\mathcal{X}) = 1$ are is essentially without loss of generality, as $\mu$ and $K$ can be rescaled. The condition $\int_{\mathcal{X}} K(x,y)\mu(\mathrm{d}x) \equiv \lambda_1$ implies that the top eigenfunction $e_1$ is equal to a constant or equivalently, that $y \mapsto K(x,y)\mu(\mathrm{d}x)$ defines a Markov kernel up to a normalizing constant.

## 3.3 Tightness of Theorems 3.1 and 3.2

**Dependence on $\|\lambda\|_2$**    An apparent weakness of Theorem 3.2 is its reliance on the unknown value $J_\epsilon^\star$, which depends on the specifics of the kernel $K$ and base measure $\mu$. Determining it is potentially highly nontrivial even for simple kernels. Slightly weakening Theorem 3.2 we obtain the following corollary, which shows that the dependence on $\|\lambda\|_2$ is tight, at least for small $\epsilon$.

**Corollary 3.5.** *Suppose $J \geq 2$ is such that $\sum_{j=2}^{J} \lambda_j^2 \geq c^2 \|\lambda\|_2^2$ for some $c \leq 1$. Then $\|\lambda\|_{2J_\epsilon^\star}$ can be replaced by $c\|\lambda\|_2$ in Theorem 3.2 whenever $\epsilon \leq c\|\lambda\|_2/(2\sqrt{J-1})$.*

**Dependence on $R$ and $\alpha$**    Due to the general nature of our lower bound constructions, it is difficult to capture the dependence on the misspecification parameter $R$. As for the probability of error $\alpha$, based on related work [16] we expect the gap of size $\sqrt{\log(1/\alpha)}$ to be a shortcoming of Theorem 3.1 and not the lower bound. Closing this gap may require a different approach, however, as tests based on empirical $L^2$ distances are known to have a sub-optimal concentration [27].

**Dependence on $\delta$**    The correct dependence on the signal rate $\delta$ is the most important question left open by our theoretical results. Any method requiring $n$ larger than a function of $\delta$ irrespective of $m$ (as in Theorem 3.1) is provably sub-optimal because taking $m \gtrsim 1/(\delta\epsilon)^2$ and $n$ large enough to estimate both $P_X, P_Y$ to within accuracy $\epsilon/10$ always suffices to reach a fixed level of total error.

## 3.4 Relation to Prior Results

In this section we discuss some connections of Theorem 3.1 to prior work. Specifically, we discuss how Theorem 3.1 recovers some known results in the literature [4; 18; 37] that are *minimax optimal*. Details omitted in this section are included in Appendix B.

**Binary Hypothesis Testing**    Suppose the two distributions $P_X, P_Y$ are *known*, we are given $m$ i.i.d. observations $Z_1, \ldots, Z_m \sim P_Z$ and our task is to decide between the hypotheses $H_0 : P_X = P_Z$ versus $H_1 : P_Y = P_Z$. Then, we may take $n = \infty, R = 0, \delta = 1$ in Theorem 3.1 to conclude that

$$m \geq c \cdot \frac{C\|\lambda\|_\infty \log(1/\alpha)}{\epsilon^2}$$

observations suffice to perform the test at total error $\alpha$.

**Two-Sample Testing**    Suppose we have two i.i.d. samples $X$ and $Y$, both of size $n$, from unknown distributions $P_X, P_Y$ respectively and our task is to decide between $H_0 : P_X = P_Y$ against $H_1 :$

$\text{MMD}(P_X, P_Y) \geq \epsilon$. We split our $Y$ sample in half resulting in $Y^{(1)}$ and $Y^{(2)}$ and form the statistic $\Psi_{\mathsf{TS}} \triangleq \Psi_{1/2}(X, Y^{(1)}, Y^{(2)}) - \Psi_{1/2}(Y^{(1)}, X, Y^{(2)})$, where $\Psi_{1/2}$ is defined in Section 2.2. Then $|\mathbb{E}\,\Psi_{\mathsf{TS}}|$ is equal to 0 under the null hypothesis and is at least $1 - 2\alpha_1$ under the alternative, where $\alpha_1$ is the target total error probability of $\Psi_{1/2}$. Taking $\alpha_1 = 5\%$, by repeated sample splitting and majority voting we may amplify the success probability to $\alpha$ provided

$$n \geq c' \frac{C\|\lambda\|_2 \log(1/\alpha)}{\epsilon^2}, \tag{7}$$

where $c' > 0$ is universal (see Appendix for details). The upper bound (7) partly recovers [37, Theorem 3 and 5] where authors show that thresholding the MMD with Gaussian kernel $G_\sigma(x, y) = \sigma^{-d} \exp(-\|x - y\|^2/\sigma^2)$ achieves the minimax optimal sample complexity $n \asymp \epsilon^{-(2\beta + d/2)/\beta}$ for the problem of two-sample testing over the class $\mathcal{P}_{\beta,d}$ of $d$-dimensional $(\beta, 2)$-Sobolev-smooth distributions (defined in Appendix B.3) under $\epsilon$-$L^2$-separation. For this, taking $\sigma \asymp \epsilon^{1/\beta}$ ensures that $\|P - Q\|_{L^2} \lesssim \text{MMD}(P, Q)$ over $P, Q \in \mathcal{P}_{\beta,d}$. Taking e.g. $\mathrm{d}\mu(x) = \exp(-\|x\|_2^2)\mathrm{d}x$ as the base measure, (7) recovers the claimed sample complexity since $\|\lambda\|_2^2 = \int G_\sigma^2(x, y)\mathrm{d}\mu(x)\mathrm{d}\mu(y) = \mathcal{O}(\sigma^{-d})$ hiding dimension dependent constants. Our result requires a bounded density with respect to a Gaussian.

**Likelihood-Free Hypothesis Testing**  By taking $\alpha \asymp R \asymp \delta = \Theta(1)$ in Theorem 3.1 we can recover many of the results in [18; 32; 33]. When $\mathcal{X}$ is finite, we can take the kernel $K(x, y) = \sum_{z \in \mathcal{X}} \mathbb{1}\{x = y = z\}$ in Theorem 3.1 to obtain the results for bounded discrete distributions (defined in Appendix B.1) which state that under $\epsilon$-TV-separation the minimax optimal sample complexity is given by $m \gtrsim 1/\epsilon^2; \min\{n, \sqrt{nm}\} \gtrsim \sqrt{|\mathcal{X}|}/\epsilon^2$. A similar kernel recovers the optimal result for the class of $\beta$-Hölder smooth densities on the hypercube $[0, 1]^d$ (see Appendix B.2).

**Curse of Dimensionality**  Using the Gaussian kernel $G_\sigma$ as for two-sample testing above, one can conclude by Theorem 3.1 that the required number of samples for (mLFHT) over the class $\mathcal{P}_{\beta,d}$ under $\epsilon$-$L^2$-separation grows at most like $\Omega\left(\left(\frac{c}{\epsilon}\right)^{\Omega(d)} \frac{1}{\delta^2}\right)$ for some $c > 1$, instead of the expected $\Omega\left(\left(\frac{c}{\epsilon\delta}\right)^{\Omega(d)}\right)$. This may be interpreted as theoretical support for the success of LFI in practice where signal and background can be rather different (cf. [5, Figures 2-3]) and the difficulty of the problem stems from the rate of signal events being small (i.e. $\epsilon \approx 1$ but $\delta \ll 1$).

# 4  Learning Kernels from Data

Given a *fixed* kernel $K$, our Theorems 3.1 and 3.2 show that the sample complexity is heavily dependent on the separation $\epsilon$ under the given MMD as well as the spectrum $\lambda = \lambda(\mu, K)$ of the kernel. Thus, to have good test performance we need to use a kernel $K$ that is well-adapted to the problem at hand. In practice, however, instead of using a fixed kernel it would be only natural to use part of the simulation sample to try to *learn* a good kernel.

In Sections 4 and 5 we report experimental results after *training* a kernel parameterized by a neural network on part of the simulation data. In particular, due to the dependence between the data and the kernel, Theorem 3.1 doesn't directly apply. Our main contribution here is showing the existence of an asymmetric simulation-experimentation trade-off (cf. Figure 1.2 and also Section 1.1) even in this realistic setting. Figure 1.2 plots the *total* number $n$ of simulations used, including those used for training, so as to provide a realistic view of the amount of data used. The experiments also illustrate that the (trained-)kernel-based statistic of Section 2.2 achieves state-of-the-art performance.

## 4.1  Proposed Training Algorithm

Consider splitting the data into three parts: $(X^{\mathsf{tr}}, Y^{\mathsf{tr}})$ is used for training (optimizing) the kernel; $(X^{\mathsf{ev}}, Y^{\mathsf{ev}})$ is used to evaluate our test statistic at test time; and $(X^{\mathsf{cal}}, Y^{\mathsf{cal}})$ is used for calibrating the distribution of the test statistic under the null hypothesis. We write $n_s = |X^s| = |Y^s|$ for $s \in \{\mathsf{tr}, \mathsf{ev}, \mathsf{cal}\}$. Given the training data $X^{\mathsf{tr}}, Y^{\mathsf{tr}}$ with empirical measures $\widehat{P}_{X^{\mathsf{tr}}}, \widehat{P}_{Y^{\mathsf{tr}}}$, we maximize the objective in

$$\widehat{J}(X^{\mathsf{tr}}, Y^{\mathsf{tr}}; K) = \frac{\text{MMD}_u^2(\widehat{P}_{X^{\mathsf{tr}}}, \widehat{P}_{Y^{\mathsf{tr}}}; K)}{\widehat{\sigma}(X^{\mathsf{tr}}, Y^{\mathsf{tr}}; K)}, \tag{8}$$

which was introduced in [48]. Here $\widehat{\sigma}^2$ is an estimator of the variance of $\text{MMD}_u^2(\widehat{P}_{X^{\mathsf{tr}}}, \widehat{P}_{Y^{\mathsf{tr}}}; K)$ and is defined in Appendix F.1.

---

**Algorithm 1** mLFHT with a learned deep kernel

---

**Input:** $(X^{\mathsf{tr}}, X^{\mathsf{ev}}, X^{\mathsf{cal}})$, $(Y^{\mathsf{tr}}, Y^{\mathsf{ev}}, Y^{\mathsf{cal}})$; parametrized kernel $K_\omega$; hyperparameters and initialization.

**# Phase 1:** *Kernel training (optimization) on $X^{\mathsf{tr}}$ and $Y^{\mathsf{tr}}$.*

   $\omega \leftarrow \arg\max_\omega^{\text{Optimizer}} \widehat{J}(X^{\mathsf{tr}}, Y^{\mathsf{tr}}; K_\omega)$;                *# maximize objective as defined in* (8)

**# Phase 2:** *Distributional calibration of test statistic (under null hypothesis).*

**for** $r = 1, 2, \ldots, k$ **do**

   $Z^{\mathsf{cal},r} \leftarrow$ sample $m$ points without replacement from $X^{\mathsf{cal}}$;

   $T_r \leftarrow \frac{1}{n_{\mathsf{ev}}m} \sum_{i,j} \left( K_\omega(Z_i^{\mathsf{cal},r}, Y_j^{\mathsf{ev}}) - K_\omega(Z_i^{\mathsf{cal},r}, X_j^{\mathsf{ev}}) \right)$;

**end for**

**# Phase 3:** *Inference with input $Z$.*

   $\widehat{T} \leftarrow \frac{1}{n_{\mathsf{ev}}m} \sum_{i,j} \left( K_\omega(Z_i, Y_j^{\mathsf{ev}}) - K_\omega(Z_i, X_j^{\mathsf{ev}}) \right)$;

**Output:** Estimated $p$-value: $\frac{1}{k} \sum_{i=1}^{k} \mathbb{1}\{\widehat{T} < T_i\}$.

---

Intuitively, the objective $J$ aims to separate $P_X$ from $P_Y$ while keeping variance low. For a heuristic justification of its use for (mLFHT) see Appendix.

In Algorithm 1 we describe the training and testing procedure, which produces unbiased $p$-values for (mLFHT) when there is no misspecification ($R = 0$ in Theorem 3.1). During training, we use the Adam optimizer [34] with stochastic batches.

**Proposition 4.1.** *When there is no misspecification ($R = 0$ in Theorem 3.1), Algorithm 1 outputs an unbiased estimate of the $p$-value that is consistent as $\min\{n_{\mathsf{cal}}, k\} \to \infty$.*

**Time complexity** Algorithm 1 runs in three separate stages: training, calibration, and inference. The first two take $O\left(\#\text{epochs} \cdot B^2 + k n_{\mathsf{ev}} m\right)$ total time, where $B$ is the batch size, whereas Phase 3 takes only $O(n_{\mathsf{ev}} m)$ time, which is generally much faster especially if $n_{\mathsf{ev}} \ll n_{\mathsf{tr}}$.

**Sample usage** Empirically, data splitting in Algorithm 1 can have non-trivial effects on performance. Instead of training the kernel on only a fraction of the data ($\{X^{\mathsf{tr}}, Y^{\mathsf{tr}}\} \cap \{X^{\mathsf{ev}}, Y^{\mathsf{ev}}\} = \varnothing$), we discovered that taking $\{X^{\mathsf{ev}}, Y^{\mathsf{ev}}\} \subseteq \{X^{\mathsf{tr}}, Y^{\mathsf{tr}}\}$ results in more efficient use of data. The stronger condition $\{X^{\mathsf{ev}}, Y^{\mathsf{ev}}\} = \{X^{\mathsf{tr}}, Y^{\mathsf{tr}}\}$ can also be applied; we take $\subseteq$ to reduce time complexity. We do, however, crucially require $X^{\mathsf{cal}}, Y^{\mathsf{cal}}$ in Phase 2 to be independently sampled ("held-out") for consistent $p$-value estimation. Finally, we remark also that splitting this way is only valid in the context of Algorithm 1. For the test (4) using the data-dependent threshold $\gamma$, one needs $\{X^{\mathsf{tr}}, Y^{\mathsf{tr}}\} \cap \{X^{\mathsf{ev}}, Y^{\mathsf{ev}}\} = \varnothing$ to estimate $\gamma$.

## 4.2 Classifier-Based Tests and Other Benchmarks

Let $\phi : \mathcal{X} \to [0, 1]$ be a classifier, assigning small values to $P_X$ and high values to $P_Y$ by minimizing cross-entropy loss of a classifier net. There are two natural test statistics based on $\phi$:

**Scheffé's Test.** The first idea, attributed to Scheffé in folklore [15, Section 6], is to take the statistic $T(Z) = \frac{1}{m} \sum_{i=1}^{m} \mathbb{1}\{\phi(Z_i) > t\}$ where $t$ is some (learn-able) threshold.

**Approximate Neyman-Pearson / Logit Methods.** If $\phi$ is trained to perfection, then $\phi(z) = \mathbb{P}(P_X|z)$ would be the likelihood and $\phi(z)/(1 - \phi(z))$ would equal precisely the likelihood ratio between $P_Y$ and $P_X$ at $z$. This motivates the use of $T(Z) = \frac{1}{m} \sum_{i=1}^{m} \log(\phi(Z_i)/(1 - \phi(Z_i)))$. See also [10].

Let us list the testing procedures that we benchmark against each other in our experiments.

1. **MMD-M**: The MMD statistic (3) using $K$ with the mixing architecture

$$K(x, y) = [(1 - \tau)G_\sigma(\varphi_\omega(x), \varphi_\omega(y)) + \tau] \cdot G_{\sigma_0}(x + \varphi'_{\omega'}(x), y + \varphi'_{\omega'}(y)).$$

   Here $G_\sigma$ is the Gaussian kernel with variance $\sigma^2$; $\varphi_\omega, \varphi'_{\omega'}$ are NN's (with parameters $\omega, \omega'$), and $\sigma, \sigma_0, \tau, \omega, \omega'$ are trained.

2. **MMD-G**: The MMD statistic (3) using the Gaussian kernel architecture $K(x, y) = G_\sigma(\varphi_\omega(x), \varphi_\omega(y))$ where $\varphi_\omega$ is the feature mapping parametrized by a trained network and $\sigma$ is a trainable parameter.

3. **MMD-O**: The MMD statistic (3) using the Gaussian Kernel $K(x, y) = G_\sigma(x, y)$ with optimized bandwidth $\sigma$. First proposed in [7; 39].

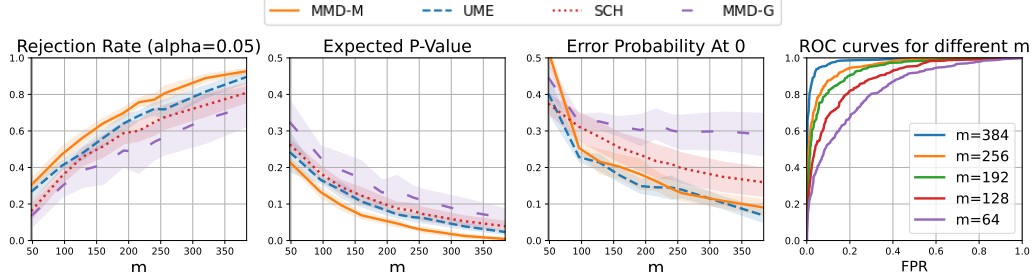

Figure 3: Empirical performance on (9) for the CIFAR detection problem when $n_{\text{tr}} = 1920$. Plots from left to right are as follows. (a) rejection rate under the alternative if test rejects whenever the estimated $p$-value is smaller than $5\%$; (b) expected $p$-value [45] under the alternative; (c) the average of type-I and II error probabilities when thresholded at 0 (different from (4), see Appendix); and (d) ROC curves for different $m$ using MMD-M and Algorithm 1. Shaded area shows the standard deviation across 10 independent runs. Missing benchmarks (thresholded MMD, MMD-O, LBI, RFM) are weaker; see Appendix for full plot.

4. **UME**: An interpretable model comparison algorithm proposed by [31], which evaluates the kernel mean embedding on a chosen "witness set".

5. **SCHE**, **LBI**: Scheffé's test and Logit Based Inference methods [10], based on a binary classifier network $\phi$ trained via cross-entropy, introduced above.

6. **RFM**: Recursive Feature Machines, a recently proposed kernel learning algorithm by [44].

### 4.3 Additive Statistics and the Thresholding Trick

Given a function $f$ (usually obtained by training) and test data $Z = (Z_1, \ldots, Z_m)$, we call a test additive if its output is obtained by thresholding $T_f(Z) \triangleq \frac{1}{m} \sum_{i=1}^{m} f(Z_i)$. We point out that all of **MMD-M/G/O**, **SCHE**, **LBI**, **UME**, **RFM** are of this form, see the Appendix for further details. Similarly to [39], we observe that any such statistic can be realized by our kernel-based approach.

**Proposition 4.2.** *The kernel-based statistic defined in* (3) *with the kernel* $K(x, y) = f(x)f(y)$ *is equal to* $T_f$ *up to a multiplicative and additive constant independent of* $Z$.

Motivated by the Scheffé's test, instead of directly thresholding the additive statistic $T_f(Z)$, we found empirically that replacing $f$ by $f_t(x) \triangleq \mathbb{1}\{f(x) > t\}$ can yield improved power. We set $t$ by maximizing an estimate of the significance under the null using a normal approximation, i.e. by solving $t_{\text{opt}} \triangleq \arg\max_t \frac{T_{f_t}(Y^{\text{opt}}) - T_{f_t}(X^{\text{opt}})}{\sqrt{T_{f_t}(X^{\text{opt}})(1 - T_{f_t}(X^{\text{opt}}))}}$, where $X^{\text{opt}}, Y^{\text{opt}}$ satisfy $\{X^{\text{opt}}, Y^{\text{opt}}\} \cap (\{X^{\text{cal}}, Y^{\text{cal}}\} \cup \{X^{\text{ev}}, Y^{\text{ev}}\}) = \varnothing$. This trick improves the performance of our tester on the Higgs dataset in Section 5.2 but not for the image detection problem in Section 5.1.

## 5 Experiments

Our code can be found at `https://github.com/Sr-11/LFI`.

### 5.1 Image Source Detection

Our first empirical study looks at the task of detecting whether images come from the CIFAR-10 [36] dataset or a SOTA generative model (DDPM) [25; 51]. While source detection is on its own interesting, it turns out that detecting whether a group of images comes from the generative model versus the real dataset can be too "easy" (see experiments in [31]). Therefore, we consider a *mixed alternative*, where the alternative hypothesis is not simply the generative model but CIFAR with planted DDPM images. Namely, our $n$ labeled images come from the following distributions:

$$P_X = \text{CIFAR}, \quad \text{and} \quad P_Y = \frac{1}{3} \cdot \text{DDPM} + \frac{2}{3} \cdot \text{CIFAR}. \tag{9}$$

The goal is to test whether the $m$ unlabeled observations $Z$ have been corrupted with $\rho$ or more fraction of DDPM images (versus uncorrupted CIFAR); this corresponds to (LFHT) (or equivalently (mLFHT) with $\delta = 1$). Figure 3 shows the performance of our approach with this mixed alternative.

**Network Architecture** With a standard deep CNN, the difference is only at the final layer: for the kernel-based tests it is a feature output; for classifiers, we add an extra linear layer to logits.

We see from Figure 3 that our kernel-based test outperforms other benchmarks at a fixed training set size $n_{tr}$. One potential cause is that MMD has an "optimization" subroutine (which it solves in closed form) as it is an IPM. This additional layer of optimization may lead to better performance at small sample sizes. The thresholding trick does not seem to improve power empirically. We omit several benchmarks from this figure for graphic presentation and they do not exhibit good separating power; see the Appendix for the complete results. The bottom plot of Figure 1.2 shows $m$ and $n$ on log-scale against the total probability of error, exhibiting the simulation-experimentation trade-off.

## 5.2 Higgs-Boson Discovery

The 2012 announcement of the Higgs boson's discovery by the ATLAS and CMS experiments [1; 9] marked a significant milestone in physics. The statistical problem inherent in the experiment is well-modeled by (mLFHT), using a signal rate $\delta$ predicted by theory and misspecification parameter $R = 0$ (as was assumed in the original discovery). We consider our algorithm's power against past studies in the physics literature [5] as measured by the *significance of discovery*. We note an important distinction from Algorithm 1 in this application.

**Estimating the Significance** In physics, the threshold for claiming a "discovery" is usually at a significance of $5\sigma$, corresponding to a $p$-value of $2.87 \times 10^{-7}$. Approximately $n_{\text{cal}} \sim (2.87)^{-1} \times 10^7$ samples would be necessary for Algorithm 1 to reach such a precision. Fortunately the distribution of the test statistic is approximated by a Gaussian customarily. We adopt this approach for our experiment hereby assuming that $m$ is large enough for the CLT to

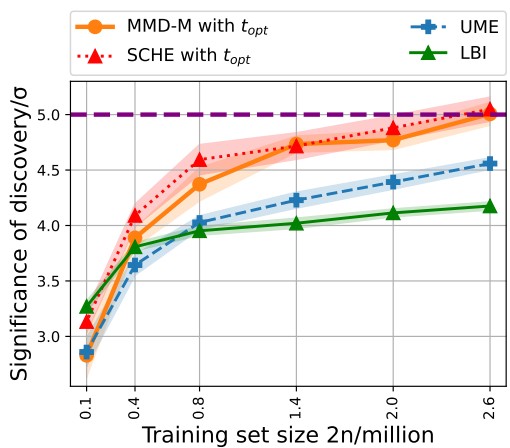

Figure 4: Expected significance of discovery on a mixture of 1000 backgrounds and 100 signals in the Higgs experiment. Shaded area shows the standard deviation over 10 independent runs. See Appendix for full plot including missing benchmarks.

apply. We use the "expected significance of discovery" as our metric [5] which, for the additive statistic $T_f = \frac{1}{m}\sum_{i=1}^{m} f(Z_i)$, is given by $\frac{\delta(T_f(Y^{\text{cal}}) - T_f(X^{\text{cal}}))}{\sqrt{\widehat{\text{var}}(f(X^{\text{cal}}))/m}}$. If the thresholding trick (Section 4.3) is applied we use the more precise Binomial tail, in which case the significance is estimated by $-\Phi^{-1}(\mathbb{P}(\text{Bin}(m, T_{f_{t_{\text{opt}}}}(X^{\text{cal}})) \geq T_{f_{t_{\text{opt}}}}(Z)))$, where $\Phi$ is the standard normal CDF.

**Newtowk Architecture** The architecture is a 6-layer feedforward net similar for all tests (kernel-based and classifiers) except for the last layer. We leave further details to the Appendix.

As can be seen in Figure 4, Scheffé's test and MMD-M with threshold $t_{\text{opt}}$ are the best methods, achieving similar performance as the algorithm of [5]; reaching the significance level of $5\sigma$ on 2.6 million simulated datapoints and a test sample made up of a mixture of 1000 backgrounds and 100 signals. The top plot of Figure 1.2 shows $m$ and $n$ on log-scale against the total probability of error through performing the test (4), exhibiting the asymmetric simulation-experimentation trade-off.

## 6 Conclusion

In this paper, we introduced (mLFHT) as a theoretical model of real-world likelihood-free signal detection problems arising in science. We proposed a kernel-based test statistic and analyzed its minimax sample complexity, obtaining both upper (Theorem 3.1) and lower bounds (Theorem 3.2) in terms of multiple problem parameters, and discussed their tightness (Section 3.3) and connections to prior work (Section 3.4). On the empirical side, we described a method for training a parametrized kernel and proposed a consistent $p$-value estimate (Algorithm 1 and Proposition 4.1). We examined the performance of our method in two experiments and found that parametrized kernels achieve state-of-the-art performance compared to relevant benchmarks from the literature. Moreover, we confirmed experimentally the existence of the asymmetric simulation-experimentation trade-off (Figure 1.2) which is suggested by minimax analysis. We defer further special cases of Theorem 3.1, all relevant proofs and experimental details to the Appendix.

## Acknowledgments and Disclosure of Funding

YP was supported in part by the National Science Foundation under Grant No CCF-2131115. Research was sponsored by the United States Air Force Research Laboratory and the Department of the Air Force Artificial Intelligence Accelerator and was accomplished under Cooperative Agreement Number FA8750-19-2-1000. The views and conclusions contained in this document are those of the authors and should not be interpreted as representing the official policies, either expressed or implied, of the Department of the Air Force or the U.S. Government. The U.S. Government is authorized to reproduce and distribute reprints for Government purposes notwithstanding any copyright notation herein.

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

# Appendix

## A  Notation

We use $A \gtrsim B, A \lesssim B, A \asymp B$ to denote $A = \Omega(B), B = \Omega(A)$ and $A = \Theta(B)$ respectively, where the hidden constants depend on untracked parameters multiplicatively.[2]

We write $\mathrm{TV}, \mathrm{KL}, \chi^2$ for total-variation, KL-divergence and $\chi^2$-divergence, respectively. We write $D(P_{Y|X} \| Q_{Y|X} | P_X) = \mathbb{E}_{X \sim P_X} D(P_{Y|X} \| Q_{Y|X})$ as the *conditional divergence* for any probability measures $P, Q$ on two variables $X, Y$ and divergence $D \in \{\mathrm{TV}, \mathrm{KL}, \chi^2\}$.

We write $\ell^p$ for the usual $\ell^p$ sequence space and $L^p$ for the usual $L^p$ space with respect to the Lebesgue measure. Both the $\ell^p$ norm and the $L^p$ norm are written as $\| \cdot \|_p$ if no ambiguity arises.

For real numbers $a, b \in \mathbb{R}$ we also write $\max\{a, b\}$ as $a \vee b$ and $\min\{a, b\}$ as $a \wedge b$.

We use $\vec{1}_d$ to denote an $d$-dimensional all 1's vector.

For an integer $k \in \mathbb{Z}^+$, we write $[k]$ as a short notation for the set $\{1, 2, \ldots, k\}$.

In the proofs of Theorem 3.1 and Theorem 3.2, we use $\overset{!}{=}$ for an equality that we are trying to prove.

## B  Applications of Theorem 3.1

Usually, minimax rates of testing are proven under separation assumptions using more traditional measures of distance such as $L^p$, where $p \in [1, \infty]$. In this section we show one example of how Theorem 3.1 can be used to recover known results, and also obtain some novel results under $L^2$-separation and $L^1$-separation.

### B.1  Bounded Discrete Distributions Under $L^2/L^1$-Separation

**Sample Complexity Upper Bounds**   Let $\mathcal{P}_{\mathsf{Db}}(k, C)$ be the set of all discrete distributions $P$ supported on $[k] = \{1, 2, \ldots, k\}$ satisfying $\max_{1 \le i \le k} p(i) \le C/k$, where $p$ is the probability mass function of $P$ (here $\sum_{i=1}^k p(k) = 1$). For distributions $P_X, P_Y, P_Z$ we shall write $p_X, p_Y, p_Z$ as their probability mass functions, respectively.

Let us apply Theorem 3.1 with underlying space $\mathcal{X} = [k]$ and measure $\mu = \frac{1}{k} \sum_{i=1}^k \delta_i$. Take the kernel $K(x, y) = \mathbb{1}\{x = y\} = \sum_{i=1}^k \mathbb{1}\{x = y = i\}$, and note that for any two distributions $P_X, P_Y$ we have

$$\mathrm{MMD}^2(P_X, P_Y) = \mathbb{E}\left[K(X, X') + K(Y, Y') - 2K(X, Y)\right] = \sum_i |p_X(i) - p_Y(i)|^2$$

where $(X, X', Y, Y') \sim P_X^{\otimes 2} \otimes P_Y^{\otimes 2}$. So the corresponding MMD is the $\ell^2$-distance on probability mass functions. Note also that $K = \sum_{i=1}^k \frac{1}{k} \left(\sqrt{k}\mathbb{1}\{x = i\}\right) \left(\sqrt{k}\mathbb{1}\{y = i\}\right)$, where $\left\{\sqrt{k}\mathbb{1}\{x = i\}\right\}_{i=1}^k$ forms an orthonormal basis of $L^2(\mu)$. So $K$ has only one nonzero eigenvalue, namely

$$\lambda_1 = \lambda_2 = \ldots = \lambda_k = 1/k,$$

of multiplicity $k$. Suppose that we observe samples $X, Y, Z$ of size $n, n, m$ from $P_X, P_Y, P_Z \in \mathcal{P}_{\mathsf{Db}}(k, C)$, where $\mathrm{MMD}(P_X, P_Y) = \sqrt{\sum_i |p_X(i) - p_Y(i)|^2} \ge \epsilon$. Plugging into Theorem 3.1 shows that:

---

[2]For example, the first equation in (10) means that there exists a constant $c$ independent of $\alpha, k, \epsilon, \delta, R$, such that $\min\{m, n\} \ge c \frac{\log(1/\alpha)(1+R)^2}{k\epsilon^2\delta^2}$.

**Proposition B.1.** *For any two $P_X, P_Y \in \mathcal{P}_{\mathsf{Db}}(k, C)$, if the $\ell^2$-distance between $p_X, p_Y$ is at least $\epsilon$, then testing (mLFHT) is possible at total error $\alpha$ using $n$ simulation samples and $m$ real data samples provided that*

$$\min\{m, n\} \gtrsim \frac{C\|\lambda\|_\infty \log(1/\alpha)(1 + R)^2}{\delta^2 \epsilon^2} \asymp \frac{\log(1/\alpha)(1 + R)^2}{k\epsilon^2\delta^2},$$

$$\min\{n, \sqrt{mn}\} \gtrsim \frac{C\|\lambda\|_2 \sqrt{\log(1/\alpha)}}{\epsilon^2 \delta} \asymp \frac{\sqrt{\log(1/\alpha)}}{\sqrt{k}\epsilon^2\delta}. \tag{10}$$

*where $R$ is defined as in the assumption $(iii)$ of Section 3.1.*

We can convert the above results to measure separation with respect to total variation (recall $\mathrm{TV}(p, q) = \frac{1}{2}\sum_i |p(i) - q(i)| = \frac{1}{2}\|p - q\|_1$) using the AM-QM inequality $\|p_X - p_Y\|_1 \leq \sqrt{k}\|p_X - p_Y\|_2$. Then, taking $R \asymp \alpha \asymp \delta = \Theta(1)$ recovers the minimax optimal results of [18; 32; 33], for LFHT over the class $\mathcal{P}_{\mathsf{Db}}$. Note that analogous results for two-sample testing follow from the above using the reduction presented in Section 3.4.

**Sample Complexity Lower Bounds**    Recall the definition of $J_\epsilon^\star$ and note that $\|\lambda\|_{2,J}^2 = \frac{\min(J-1, k)}{k^2}$ for all $J \geq 2$. By Corollary 3.5 we see that $J_\epsilon^\star \gtrsim k$ as soon as $\epsilon \lesssim 1/k$. Thus, for $\epsilon \lesssim 1/k$ the necessity of

$$m \gtrsim \frac{\log(1/\alpha)}{k\epsilon^2\delta^2}, \; n \gtrsim \frac{\sqrt{\log(1/\alpha)}}{\sqrt{k}\epsilon^2} \text{ and } m + \sqrt{mn} \gtrsim \frac{\sqrt{\log(1/\alpha)}}{\sqrt{k}\epsilon^2\delta} \tag{11}$$

follows by Theorem 3.2. Here it is crucial to note that when $\delta = \Theta(1)$, we have

$$m + \sqrt{mn} \gtrsim \frac{\sqrt{\log(1/\alpha)}}{\sqrt{k}\epsilon^2} \text{ and } n \gtrsim \frac{\sqrt{\log(1/\alpha)}}{\sqrt{k}\epsilon^2} \iff \sqrt{mn} \gtrsim \frac{\sqrt{\log(1/\alpha)}}{\sqrt{k}\epsilon^2} \text{ and } n \gtrsim \frac{\sqrt{\log(1/\alpha)}}{\sqrt{k}\epsilon^2}$$

and hence the upper bound (10) meets with the lower bound (11) provided $R \asymp \delta = \Theta(1)$. Once again, setting $R \asymp \delta \asymp \alpha = \Theta(1)$ we the optimal lower bounds recovering the results of [18] (in the regime $\epsilon \lesssim 1/k$). In short we can also recover the following result for LFHT.

**Proposition B.2** ([18, Theorem 1, adapted]).  *On the class $P_{\mathsf{Db}}(k, C)$, using $n$ simulation samples and $m$ real data samples, if*

$$n \gtrsim \frac{1}{\sqrt{k}\epsilon^2}, \quad m \gtrsim \frac{1}{k\epsilon^2}, \quad \sqrt{mn} \gtrsim \frac{1}{\sqrt{k}\epsilon^2}, \tag{12}$$

*then for any two distributions $P_X, P_Y \in \mathcal{P}_{\mathsf{Db}}(k, C)$ with $\|p_X - p_Y\|_2 \geq \epsilon$, testing (LFHT) is possible with a total error of $1\%$. Conversely, to ensure the existence of a procedure that can test (LFHT) with a total error of $1\%$ for any $P_X, P_Y \in \mathcal{P}_{\mathsf{Db}}(k, C)$ with $\|p_X - p_Y\|_2 \geq \epsilon$, the number of observations $(n, m)$ must satisfy*

$$n \gtrsim \frac{1}{\sqrt{k}\epsilon^2}, \quad m \gtrsim \frac{1}{k\epsilon^2}, \quad \sqrt{mn} \gtrsim \frac{1}{\sqrt{k}\epsilon^2}. \tag{13}$$

*The implied constants in (12) and (13) do not depend on $k$ and $\epsilon$, but may differ.*

## B.2   $\beta$-Hölder Smooth Densities on $[0, 1]^d$ Under $L^2/L^1$-Separation

**Sample Complexity Upper Bounds**    Let $\mathcal{P}_{\mathsf{H}}(\beta, d, C)$ be the set of all distributions on $[0, 1]^d$ with $\beta$-Hölder smooth Lebesgue-density $p$ satisfying $\|p\|_{\mathcal{C}^\beta} \leq C$ for some constant $C > 1$, where

$$\|p\|_{\mathcal{C}^\beta} \triangleq \max_{0 \leq |\alpha| \leq \lceil \beta - 1 \rceil} \|f^{(\alpha)}\|_\infty + \sup_{x \neq y \in [0,1]^d, |\alpha| = \lceil \beta - 1 \rceil} \frac{|f^{(\alpha)}(x) - f^{(\alpha)}(y)|}{\|x - y\|_2^{\beta - \lceil \beta - 1 \rceil}},$$

where $\lceil \beta - 1 \rceil$ is the largest integer strictly smaller than $\beta$ and $|\alpha| = \sum_i \alpha_i$ is the norm of a multi-index $\alpha \in \mathbb{N}^d$. Abusing notation, we also use $\mathcal{P}_{\mathsf{H}}(\beta, d, C)$ to denote the set of all corresponding density functions.

We take $K(x, y) = \sum_j \mathbb{1}\{x, y \in B_j\}$, where $\{B_j\}_{j \in [\kappa]^d}$ is the $j$'th cell of the regular grid of size $\kappa^d$ on $[0, 1]^d$, i.e., $B_j = [(j - \vec{1}_d)/\kappa, j/\kappa]$ for $j \in [\kappa]^d$. Clearly there are $\kappa^d$ nonzero eigenvalues, each equal to 1. The following approximation result is due to Ingster [28], see also [4, Lemma 7.2].

**Lemma B.3.** *Let $f, g \in \mathcal{P}_{\mathsf{H}}(\beta, d, C)$ with $\|f - g\|_2 \geq \epsilon$. Then, there exist constants $c, c'$ independent of $\epsilon$ such that for any $\kappa \geq c\epsilon^{-1/\beta}$,*

$$\mathrm{MMD}(f, g) \geq c'\|f - g\|_2.$$

Now, suppose that we have samples $X, Y, Z$ of size $n, n, m$ from $P_X, P_Y, P_Z \in \mathcal{P}_{\mathsf{H}}(\beta, d, C)$ with densities $p_X, p_Y, p_Z$ such that $\|p_X - p_Y\|_2 \geq \epsilon$. Then, Theorem 3.1 combined with Lemma B.3 and the choice $\kappa \asymp \epsilon^{-1/\beta}$ shows that

**Proposition B.4.** *Testing* (mLFHT) *on $\mathcal{P}_{\mathsf{H}}(\beta, d, C)$ at total error $\alpha$ using $n$ simulation and $m$ real data samples is possible provided*

$$\min\{m, n\} \gtrsim \frac{C\|\lambda\|_\infty \log(1/\alpha)(1 + R)^2}{\delta^2 \epsilon^2} \asymp \frac{\log(1/\alpha)(1 + R)^2}{\delta^2 \epsilon^2},$$

$$\min\{n, \sqrt{nm}\} \gtrsim \frac{C\|\lambda\|_2 \sqrt{\log(1/\alpha)}}{\epsilon^2 \delta} \asymp \frac{\sqrt{\log(1/\alpha)}}{\epsilon^{(2\beta + d/2)/\beta} \delta},$$

*where $\epsilon$ is an $L^2$-distance lower bound between $P_X, P_Y$ and $R$ is defined as in the assumption $(iii)$ of Section 3.1.*

Setting $R \asymp \alpha \asymp \delta = \Theta(1)$ recovers the optimal results of [18] for the class $\mathcal{P}_{\mathsf{H}}$. Once again, identical results under $L^1$ separation follow from Jensen's inequality $\|\cdot\|_{L^1([0,1]^d)} \leq \|\cdot\|_{L^2([0,1]^d)}$. Note that analogous results for two-sample testing follow from the above using the reduction presented in Section 3.4.

**Sample Complexity Lower Bounds**    The kernel defined in the previous paragraph is not suitable for constructing lower bounds over the class $\mathcal{P}_{\mathsf{H}}$ because its eigenfunctions do not necessarily lie in $\mathcal{P}_{\mathsf{H}}$. It would be possible to consider a different kernel that is more adapted to this problem/class but we do not pursue this here.

## B.3    $(\beta, 2)$-Sobolev Smooth Densities on $\mathbb{R}^d$ Under $L^2$-Separation

**Sample Complexity Upper Bounds**    Let $\mathcal{P}_{\mathsf{S}}(\beta, d, C)$ be the class of distributions that are supported on $\mathbb{R}^d$ and whose Lebesgue density $p$ satisfies $\|p\|_{\beta,2} \leq C$, where

$$\|p\|_{\beta,2} \triangleq \left\|(1 + \|\cdot\|)^\beta \mathcal{F}[p]\right\|_2 \tag{14}$$

and $\mathcal{F}$ denotes the Fourier transform. Again, abusing notation, we write $\mathcal{P}_{\mathsf{S}}(\beta, d, C)$ both as the set of distributions and the set of density functions.

We take the Gaussian kernel $G_\sigma(x, y) = \sigma^{-d} \exp(-\|x - y\|_2^2/\sigma^2)$ on $\mathcal{X} = \mathbb{R}^d$ with base measure $\mathrm{d}\mu(x) = \exp(-x^2)\mathrm{d}x$. In [37] the authors showed that the two-sample test that thresholds the Gaussian MMD with appropriately chosen variance $\sigma^2$ achieves the minimax optimal sample complexity over $\mathcal{P}_{\mathsf{S}}$, when separation is measured by $L^2$. A key ingredient in their proof is the following inequality.

**Lemma B.5** ([37, Lemma 5]). *Let $f, g \in \mathcal{P}_{\mathsf{S}}(\beta, d, C)$ with $\|f - g\|_2 \geq \epsilon$. Then, there exist constants $c, c'$ independent of $\epsilon$ such that for any $\sigma \leq c\,\epsilon^{1/\beta}$, we have*

$$\mathrm{MMD}(f, g) \geq c'\|f - g\|_2.$$

Now, suppose that we have samples $X, Y, Z$ of sizes $n, n, m$ from $P_X, P_Y, P_Z \in \mathcal{P}_{\mathsf{S}}(\beta, d, C)$ for some constant $C$ with densities $p_X, p_Y, p_Y$ satisfying $\|p_X - p_Y\|_2 \geq \epsilon$.

Note that the heat-semigroup is an $L^2$-contraction ($\|\lambda\|_\infty \leq 1$) and that

$$\|\lambda\|_2^2 = \int G_\sigma(x, y)^2 \mathrm{d}\mu(x)\mathrm{d}\mu(y) \asymp \sigma^{-d}$$

up to constants depending on the dimension. Theorem 3.1 combined with Lemma B.5 and a choice $\sigma \asymp \epsilon^{1/\beta}$ yields the following result.

**Proposition B.6.** *Testing* (mLFHT) *over the class* $\mathcal{P}_S$ *with total error* $\alpha$ *is possible provided*

$$\min\{m, n\} \gtrsim \frac{C\|\lambda\|_\infty \log(1/\alpha)(1 + R)^2}{\delta^2 \epsilon^2} \asymp \frac{\log(1/\alpha)(1 + R)^2}{\delta^2 \epsilon^2}$$

$$\min\{n, \sqrt{nm}\} \gtrsim \frac{C\|\lambda\|_2 \sqrt{\log(1/\alpha)}}{\epsilon^2 \delta} \asymp \frac{\sqrt{\log(1/\alpha)}}{\epsilon^{(2\beta + d/2)/\beta} \delta},$$

*where $\epsilon$ is the lower bound on the $L^2$-distance between $P_X, P_Y$ and $R$ is defined as in the assumption* $(iii)$ *of Section* 3.1.

Taking $R \asymp \delta \asymp \alpha = \Theta(1)$ above, we obtain new results for LFHT and using the reduction from two-sample testing given in Section 3.4 we partly recover [37, Theorem 5]. Only partly, because the above requires bounded density with respect to our base measure $\mathrm{d}\mu(x) = \exp(-x^2)\mathrm{d}x$.

**Sample Complexity Lower Bounds**    Note that our lower bound Theorem 3.2 doesn't apply because the top eigenfunction of the Gaussian kernel is not constant. Once again, a more careful choice of the base measure (or kernel) might lead to a more suitable argument for the lower bound. We leave such pursuit as open.

## C   Black-box Boosting of Success Probability

In this section we briefly describe how upper bounds on the minimax sample complexity in the constant error probability regime ($\alpha = \Theta(1)$) can be used to obtain the dependence $\log(1/\alpha)$ in the small error probability regime ($\alpha = o(1)$). We will argue abstractly in a way that applies to the setting of Theorem 3.1.

Suppose that from some distributions $P_1, P_2, \ldots, P_k$ we take samples $X^1, X^2, \ldots, X^k$ of size $n_1, n_2, \ldots, n_k$ respectively and are able to decide between two hypotheses $H_0$ and $H_1$ (fixed but arbitrary) with total error probability at most $1/3$. Call this test as $\Psi(X^1, \ldots, X^k) \in \{0, 1\}$, so that

$$\mathbb{P}(\Psi(X^1, \ldots, X^k) = 0|H_0) \geq 2/3 \qquad \text{and} \qquad \mathbb{P}(\Psi(X^1, \ldots, X^k) = 1|H_1) \geq 2/3.$$

Now, to each an error of $o(1)$, instead, we take $18n_1 \log(2/\alpha), \ldots, 18n_k \log(2/\alpha)$ observations from $P_1$ through $P_k$, and split each sample into $18 \log(2/\alpha)$ equal sized batches $\{X^{i,j}\}_{i \in [k], j \in [18 \log(2/\alpha)]}$. Here $18 \log(2/\alpha)$ is assumed to be an integer without loss of generality. The split samples form $18 \log(2/\alpha)$ independent binary random variables

$$A_j \triangleq \Psi(X^{1,j}, \ldots, X^{k,j})$$

for $j = 1, 2, \ldots, 18 \log(2/\alpha)$. We claim that the majority voting test

$$\Psi_\alpha(\{X^{i,j}\}_{i,j}) = \begin{cases} 1 & \text{if } \bar{A} \geq 1/2 \\ 0 & \text{otherwise} \end{cases}$$

tests $H_0$ against $H_1$ with total probability of error at most $\alpha$, where

$$\bar{A} \triangleq \frac{1}{18 \log(2/\alpha)} \sum_{j=1}^{18 \log(2/\alpha)} A_j.$$

Indeed, by Hoeffding's inequality, we have

$$\mathbb{P}\left(\bar{A} \geq 1/2 \,\middle|\, H_0\right) \leq \alpha/2$$
$$\mathbb{P}\left(\bar{A} \leq 1/2 \,\middle|\, H_1\right) \leq \alpha/2.$$

Therefore, in the remainder of our upper bound proofs, we only focus on achieving a constant probability of error ($\alpha = \Theta(1)$) as the logarithmic dependence follows by the above.

*Remark* C.1.  As mentioned in the discussion succeeding Corollary 3.5, we do conjecture the *tight* dependence in the upper bound to be $\sqrt{\log(\alpha^{-1})}$ instead of $\log(\alpha^{-1})$ shown by this method.

# D    Proof of Theorem 3.1

## D.1    Notation and Technical Tools

We use the expansion

$$K(x, y) = \sum_{\ell} \lambda_\ell e_\ell(x) e_\ell(y)$$

extensively, where $\lambda \triangleq (\lambda_1, \lambda_2, \dots)$ are $K$'s eigenvalues (regarded as an integral operator on $L^2(\mu)$) in non-increasing order and $e_1, e_2, \dots$ are the corresponding eigenfunctions forming an orthonormal basis for $L^2(\mu)$, and convergence is to be understood in $L^2(\mu)$. We use the notation $\langle \cdot \rangle \triangleq \int \cdot \, d\mu$. For all $u \in L^2(\mu)$ we define

$$u_\ell \triangleq \langle u e_\ell \rangle, \quad u_{\ell\ell'} \triangleq \langle u e_\ell e_{\ell'} \rangle, \quad \ell = 1, 2, \dots$$

and consequently $u = \sum_\ell u_\ell e_\ell$. We also define

$$K[u](\cdot) \triangleq \int K(t, \cdot) u(t) \mu(dt) = \sum_\ell \lambda_\ell u_\ell e_\ell(\cdot),$$

where the second equality follows from the orthonormality of $\{e_\ell\}_{\ell=1}^\infty$. Note that the RKHS embedding satisfies $\theta_u \triangleq \int K(x, \cdot) u(x) d\mu(x) = K[u]$. Now, for $P_X$ we write

$$x_\ell \triangleq (p_X)_\ell = \langle p_X e_\ell \rangle, \quad x_{\ell\ell'} \triangleq (p_X)_{\ell\ell'} = \langle p_X e_\ell e_{\ell'} \rangle, \quad \ell, \ell' = 1, 2, \dots$$

where $p_X$ is the $\mu$-density of $P_X$. The similar notations also apply to $P_Y, P_Z$. The following identities will be very useful in our proofs.

**Lemma D.1.** *For each identity below, let $f, g, h \in L^2(\mu)$ be such that the quantity is well defined. Then,*

$$\|\theta_f\|_{\mathcal{H}_K}^2 = \sum_\ell \lambda_\ell f_\ell^2 \tag{15}$$

$$\mathrm{MMD}^2(f, g) = \sum_\ell \lambda_\ell (f_\ell - g_\ell)^2 \tag{16}$$

$$\|K[f]\|_2^2 = \sum_\ell \lambda_\ell^2 f_\ell^2 \tag{17}$$

$$\sum_\ell \lambda_\ell f_\ell g_\ell = \langle f K[g] \rangle = \langle K[f] g \rangle \tag{18}$$

$$\sum_{\ell\ell'} \lambda_\ell \lambda_{\ell'} h_{\ell\ell'} f_\ell g_{\ell'} = \langle h K[f] K[g] \rangle \tag{19}$$

$$\sum_{\ell\ell'} \lambda_\ell \lambda_{\ell'} g_{\ell\ell'} f_{\ell\ell'} = \sum_\ell \lambda_\ell \langle f e_\ell K[g e_\ell] \rangle. \tag{20}$$

*Suppose that $f, g$ are probability densities with respect to $\mu$ that are bounded by $C$. Then*

$$0 \leq \sum_{\ell\ell'} \lambda_\ell \lambda_{\ell'} g_{\ell\ell'} f_{\ell\ell'} \leq C^2 \|\lambda\|_2^2. \tag{21}$$

*Proof.* We prove each claim, starting with (15). Clearly

$$
\begin{aligned}
\|\theta_f\|_{\mathcal{H}_K}^2 &= \|K[f]\|_{\mathcal{H}_K}^2 \\
&= \left\| \int K(x, \cdot) f(x) d\mu(x) \right\|_{\mathcal{H}_K}^2 \\
&= \iint \langle K(x, \cdot), K(y, \cdot) \rangle_{\mathcal{H}_K} f(x) f(y) d\mu(x) d\mu(y) \\
&= \iint K(x, y) f(x) f(y) d\mu(x) d\mu(y) \\
&= \sum_\ell \lambda_\ell f_\ell^2
\end{aligned}
$$

as required. The second claim (16) follows immediately from (15) by definition. For (17) by orthogonality we have

$$\|K[f]\|_2^2 = \|\sum_\ell \lambda_\ell f_\ell e_\ell\|_2^2$$
$$= \sum_\ell \lambda_\ell^2 f_\ell^2.$$

For (18) by the definition of $K[\cdot]$ we have

$$\sum_\ell \lambda_\ell f_\ell g_\ell = \left\langle \left(\sum_\ell \lambda_\ell f_\ell e_\ell\right) g \right\rangle$$
$$= \langle K[f]g\rangle.$$

For (19) we can write

$$\sum_{\ell\ell'} \lambda_\ell \lambda_{\ell'} h_{\ell\ell'} f_\ell g_{\ell'} = \sum_\ell \lambda_\ell f_\ell \left\langle \left(\sum_{\ell'} \lambda_{\ell'} g_{\ell'} e_{\ell'}\right) h e_\ell \right\rangle$$
$$= \sum_\ell \lambda_\ell f_\ell \langle K[g] h e_\ell \rangle$$
$$= \langle K[g] h K[f]\rangle.$$

Finally, for (20) we have

$$\sum_{\ell\ell'} \lambda_\ell \lambda_{\ell'} f_{\ell\ell'} g_{\ell\ell'} = \sum_\ell \lambda_\ell \left\langle \left(\sum_{\ell'} \lambda_{\ell'} g_{\ell\ell'} e_{\ell'}\right) f e_\ell \right\rangle$$
$$= \sum_\ell \lambda_\ell \langle K[g e_\ell] f e_\ell \rangle.$$

Suppose now that $f, g$ are probability densities with respect to $\mu$ that are bounded by $C > 0$. Let $X, Y$ be independent random variables following the densities $f, g$. Then

$$\sum_{\ell\ell'} \lambda_\ell \lambda_{\ell'} f_{\ell\ell'} g_{\ell\ell'} = \mathbb{E}\left[\left(\sum_\ell \lambda_\ell e_\ell(X) e_\ell(Y)\right)^2\right]$$
$$\leq C^2 \int_{\mathcal{X}} \int_{\mathcal{X}} \left(\sum_\ell \lambda_\ell e_\ell(x) e_\ell(y)\right)^2 \mathrm{d}\mu(x)\mathrm{d}\mu(y)$$
$$= C^2 \|\lambda\|_2^2$$

as claimed, where we used that the $e_\ell$ are orthonormal. $\qquad\square$

## D.2 Mean and Variance Computation

We take $\pi = \delta/2$. Our statistic reads

$$-T(X,Y,Z) + \gamma(X,Y,\pi) = \langle \theta_{\widehat{P}_Z} - (\bar{\pi}\theta_{\widehat{P}_X} + \pi\theta_{\widehat{P}_Y}), \theta_{\widehat{P}_X} - \theta_{\widehat{P}_Y}\rangle_{u,\mathcal{H}_K}$$

$$= \frac{1}{nm}\underbrace{\sum_{ij} k(X_i, Z_j)}_{\text{I}} - \frac{1}{nm}\underbrace{\sum_{ij} k(Y_i, Z_j)}_{\text{II}} - \frac{2\bar{\pi}}{n(n-1)}\underbrace{\sum_{i<i'} k(X_i, X_{i'})}_{\text{III}}$$

$$+ \frac{2\pi}{n(n-1)}\underbrace{\sum_{i<i'} k(Y_i, Y_{i'})}_{\text{IV}} + \frac{\bar{\pi} - \pi}{n^2}\underbrace{\sum_{ij} k(X_i, Y_j)}_{\text{V}}.$$

Recall that $\nu = \arg\min_{\nu' \in \mathbb{R}} \text{MMD}(P_Z, \bar{\nu}' P_X + \nu' P_Y)$. Let us write $z = \bar{\nu}x + \nu y + r$ for $1 - \bar{\nu} = \nu$, where the residual term is denoted as $r \in L^2(\mu)$. Let $\theta_r = \int r(t) K(t, \cdot) \mu(\mathrm{d}t)$ be the mean embedding of $r$. Under both hypotheses we assume that $\|\theta_r\|_{\mathcal{H}_K} \leq R \cdot \text{MMD}(P_X, P_Y)$, moreover $\langle \theta_r, \theta_{P_Y} - \theta_{P_X} \rangle_{\mathcal{H}_K} = 0$ by the definition of $\nu$. We look at each of the $5 + \binom{5}{2} = 15$ terms of the variance separately.

$$\text{var}(\mathsf{I}) = \sum_{\ell\ell'} \lambda_\ell \lambda_{\ell'} \Big\{ n(n-1)m(z_{\ell\ell'} - z_\ell z_{\ell'})x_\ell x_{\ell'} + nm(m-1)(x_{\ell\ell'} - x_\ell x_{\ell'})z_\ell z_{\ell'}$$
$$+ nm(x_{\ell\ell'} z_{\ell\ell'} - x_\ell x_{\ell'} z_\ell z_{\ell'}) \Big\}$$

$$\text{var}(\mathsf{II}) = \sum_{\ell\ell'} \lambda_\ell \lambda_{\ell'} \Big\{ n(n-1)m(z_{\ell\ell'} - z_\ell z_{\ell'})y_\ell y_{\ell'} + nm(m-1)(y_{\ell\ell'} - y_\ell y_{\ell'})z_\ell z_{\ell'}$$
$$+ nm(y_{\ell\ell'} z_{\ell\ell'} - y_\ell y_{\ell'} z_\ell z_{\ell'}) \Big\}$$

$$\text{var}(\mathsf{III}) = \sum_{\ell\ell'} \lambda_\ell \lambda_{\ell'} \Big\{ \binom{n}{2}(x_{\ell\ell'}^2 - x_\ell^2 x_{\ell'}^2) + (\binom{n}{2}^2 - \binom{n}{2} - \binom{4}{2}\binom{n}{4})(x_{\ell\ell'} - x_\ell x_{\ell'})x_\ell x_{\ell'} \Big\}$$

$$\text{var}(\mathsf{IV}) = \sum_{\ell\ell'} \lambda_\ell \lambda_{\ell'} \Big\{ \binom{n}{2}(y_{\ell\ell'}^2 - y_\ell^2 y_{\ell'}^2) + (\binom{n}{2}^2 - \binom{n}{2} - \binom{4}{2}\binom{n}{4})(y_{\ell\ell'} - y_\ell y_{\ell'})y_\ell y_{\ell'} \Big\}$$

$$\text{var}(\mathsf{V}) = \sum_{\ell\ell'} \lambda_\ell \lambda_{\ell'} \Big\{ n^2(n-1)(y_{\ell\ell'} - y_\ell y_{\ell'})x_\ell x_{\ell'} + n^2(n-1)(x_{\ell\ell'} - x_\ell x_{\ell'})y_\ell y_{\ell'}$$
$$+ n^2(x_{\ell\ell'} y_{\ell\ell'} - x_\ell x_{\ell'} y_\ell y_{\ell'}) \Big\}$$

For the cross terms we obtain

$$\text{cov}(\mathsf{I}, \mathsf{II}) = \sum_{\ell\ell'} \lambda_\ell \lambda_{\ell'} n^2 m(z_{\ell\ell'} - z_\ell z_{\ell'})x_\ell y_{\ell'}$$

$$\text{cov}(\mathsf{I}, \mathsf{III}) = \sum_{\ell\ell'} \lambda_\ell \lambda_{\ell'} n(n-1)m(x_{\ell\ell'} - x_\ell x_{\ell'})z_\ell x_{\ell'}$$

$$\text{cov}(\mathsf{I}, \mathsf{IV}) = 0$$

$$\text{cov}(\mathsf{I}, \mathsf{V}) = \sum_{\ell\ell'} \lambda_\ell \lambda_{\ell'} n^2 m(x_{\ell\ell'} - x_\ell x_{\ell'})z_\ell y_{\ell'}$$

$$\text{cov}(\mathsf{II}, \mathsf{III}) = 0$$

$$\text{cov}(\mathsf{II}, \mathsf{IV}) = \sum_{\ell\ell'} \lambda_\ell \lambda_{\ell'} n(n-1)m(y_{\ell\ell'} - y_\ell y_{\ell'})z_\ell y_{\ell'}$$

$$\text{cov}(\mathsf{II}, \mathsf{V}) = \sum_{\ell\ell'} \lambda_\ell \lambda_{\ell'} n^2 m(y_{\ell\ell'} - y_\ell y_{\ell'})z_\ell x_{\ell'}$$

$$\text{cov}(\mathsf{III}, \mathsf{IV}) = 0$$

$$\text{cov}(\mathsf{III}, \mathsf{V}) = \sum_{\ell\ell'} \lambda_\ell \lambda_{\ell'} n^2(n-1)(x_{\ell\ell'} - x_\ell x_{\ell'})x_\ell y_{\ell'}$$

$$\text{cov}(\mathsf{IV}, \mathsf{V}) = \sum_{\ell\ell'} \lambda_\ell \lambda_{\ell'} n^2(n-1)(y_{\ell\ell'} - y_\ell y_{\ell'})y_\ell x_{\ell'}.$$

Note that $\binom{n}{2}^2 - \binom{n}{2} - \binom{n}{2}\binom{n}{4} = n(n-1)^2 - n(n-1)$. Collecting terms, and simplifying, we get the coefficient of the $\frac{1}{n}$ term:

$$\text{Coef}\left(\frac{1}{n}\right) = \sum_{\ell,\ell'} \lambda_\ell \lambda_{\ell'} \Bigg( \underbrace{(x_{\ell\ell'} - x_\ell x_{\ell'})z_\ell z_{\ell'}}_{\text{var(I)}} + \underbrace{(y_{\ell\ell'} - y_\ell y_{\ell'})z_\ell z_{\ell'}}_{\text{var(II)}} + \underbrace{4\bar{\pi}^2(x_{\ell\ell'} - x_\ell x_{\ell'})x_\ell x_{\ell'}}_{\text{var(III)}}$$

$$+ \underbrace{4\pi^2(y_{\ell\ell'} - y_\ell y_{\ell'})y_\ell y_{\ell'}}_{\text{var(IV)}} + \underbrace{(\bar{\pi}-\pi)^2(y_{\ell\ell'} - y_\ell y_{\ell'})x_\ell x_{\ell'} + (\bar{\pi}-\pi)^2(x_{\ell\ell'} - x_\ell x_{\ell'})y_\ell y_{\ell'}}_{\text{var(V)}}$$

$$- \underbrace{4\bar{\pi}(x_{\ell\ell'} - x_\ell x_{\ell'})z_\ell x_{\ell'}}_{\text{cov(I,III)}} + \underbrace{2(\bar{\pi}-\pi)(x_{\ell\ell'} - x_\ell x_{\ell'})z_\ell y_{\ell'}}_{\text{cov(I,V)}}$$

$$- \underbrace{4\pi(y_{\ell\ell'} - y_\ell y_{\ell'})z_\ell y_{\ell'}}_{\text{cov(II,IV)}} - \underbrace{2(\bar{\pi}-\pi)(y_{\ell\ell'} - y_\ell y_{\ell'})z_\ell x_{\ell'}}_{\text{cov(II,V)}}$$

$$- \underbrace{4\bar{\pi}(\bar{\pi}-\pi)(x_{\ell\ell'} - x_\ell x_{\ell'})x_\ell y_{\ell'}}_{\text{cov(III,V)}} + \underbrace{4\pi(\bar{\pi}-\pi)(y_{\ell\ell'} - y_\ell y_{\ell'})y_\ell x_{\ell'}}_{\text{cov(IV,V)}} \Bigg).$$

After expanding $z_\ell$ as $z_\ell = \bar{\nu}x_\ell + \nu y_\ell + r_\ell$, we split the calculation into multiple parts to simplify it. First, we focus on terms that are multiplied by $(x_{\ell\ell'} - x_\ell x_{\ell'})$ and do not contain $r_\ell$ or $r_{\ell'}$. Using Lemma D.1 extensively and the fact that $\bar{\pi} = 1 - \pi, \bar{\nu} = 1 - \nu$, we find that the sum of these terms equals

$$\bar{\nu}^2\langle xK[x]^2\rangle + \nu^2\langle xK[y]^2\rangle + 2\bar{\nu}\nu\langle xK[x]K[y]\rangle - \bar{\nu}^2\langle xK[x]\rangle^2 - \nu^2\langle xK[y]\rangle^2 - 2\bar{\nu}\nu\langle xK[x]\rangle\langle xK[y]\rangle$$

$$+ 4\bar{\pi}^2\langle xK[x]^2\rangle - 4\bar{\pi}^2\langle xK[x]\rangle^2 + (\bar{\pi}-\pi)^2\langle xK[y]^2\rangle - (\bar{\pi}-\pi)^2\langle xK[y]\rangle^2$$

$$- 4\bar{\pi}\bar{\nu}\langle xK[x]^2\rangle - 4\bar{\pi}\nu\langle xK[x]K[y]\rangle + 4\bar{\pi}\bar{\nu}\langle xK[x]\rangle^2 + 4\bar{\pi}\nu\langle xK[x]\rangle\langle xK[y]\rangle$$

$$+ 2(\bar{\pi}-\pi)\bar{\nu}\langle xK[x]K[y]\rangle + 2(\bar{\pi}-\pi)\nu\langle xK[y]^2\rangle - 2(\bar{\pi}-\pi)\bar{\nu}\langle xK[x]\rangle\langle xK[y]\rangle - 2(\bar{\pi}-\pi)\nu\langle xK[y]\rangle^2$$

$$- 4\bar{\pi}(\bar{\pi}-\pi)\langle xK[x]K[y]\rangle + 4\bar{\pi}(\bar{\pi}-\pi)\langle xK[x]\rangle\langle xK[y]\rangle$$

$$= (\bar{\nu} - 2\bar{\pi})^2\Big(\langle xK[x-y]^2\rangle - \langle xK[x-y]\rangle^2\Big)$$

$$\le C\|\lambda\|_\infty \text{MMD}^2(P_X, P_Y).$$

Similarly, the terms involving $(y_{\ell\ell'} - y_\ell y_{\ell'})$ but not $r_\ell$ or $r_{\ell'}$ sum up to the quantity

$$(\nu - 2\pi)^2\Big(\langle yK[x-y]^2\rangle - \langle yK[x-y]\rangle^2\Big) \le C\|\lambda\|_\infty \text{MMD}^2(P_X, P_Y).$$

Next, collecting the terms involving both $(x_{\ell\ell'} - x_\ell x_{\ell'})$ and $r_\ell$ or $r_{\ell'}$ we get

$$2\bar{\nu}\langle xK[r]K[x]\rangle + 2\nu\langle xK[r]K[y]\rangle + \langle xK[r]^2\rangle - 2\bar{\nu}\langle xK[x]\rangle\langle xK[r]\rangle - 2\nu\langle xK[y]\rangle\langle xK[r]\rangle - \langle xK[r]\rangle^2$$

$$- 4\bar{\pi}\langle xK[x]K[r]\rangle + 4\bar{\pi}\langle xK[x]\rangle\langle xK[r]\rangle$$

$$+ 2(\bar{\pi}-\pi)\langle xK[y]K[r]\rangle - 2(\bar{\pi}-\pi)\langle xK[y]\rangle\langle xK[r]\rangle$$

$$= 2(\bar{\nu} - 2\bar{\pi})\Big(\langle xK[r]K[x-y]\rangle - \langle xK[r]\rangle\langle xK[x-y]\rangle\Big) + \langle xK[r]^2\rangle - \langle xK[r]\rangle^2$$

$$\lesssim C\|\lambda\|_\infty(R + R^2)\text{MMD}^2(P_X, P_Y).$$

Finally, collecting the terms involving both $(y_{\ell\ell'} - y_\ell y_{\ell'})$ and $r_\ell$ or $r_{\ell'}$ we get

$$2(\nu - 2\pi)\Big(\langle yK[r]K[y-x]\rangle - \langle yK[r]\rangle\langle yK[y-x]\rangle\Big) + \langle yK[r]^2\rangle - \langle yK[r]\rangle^2$$

$$\lesssim C\|\lambda\|_\infty(R + R^2)\text{MMD}^2(P_X, P_Y).$$

Similarly we get

$$\text{Coef}\left(\frac{1}{m}\right) = \sum_{\ell\ell'} \lambda_\ell \lambda_{\ell'} \Bigg( \underbrace{(z_{\ell\ell'} - z_\ell z_{\ell'})x_\ell x_{\ell'}}_{\text{var(I)}} + \underbrace{(z_{\ell\ell'} - z_\ell z_{\ell'})y_\ell y_{\ell'}}_{\text{var(I)}} + \underbrace{2(z_{\ell\ell'} - z_\ell z_{\ell'})x_\ell y_{\ell'}}_{\text{cov(I,II)}} \Bigg)$$

$$= \langle zK[x-y]^2\rangle - \langle zK[x-y]\rangle^2$$

$$\lesssim C\|\lambda\|_\infty \text{MMD}^2(P_X, P_Y).$$

The remaining coefficients don't rely on subtle cancellations, and simple bounds yield

$$
\text{Coef}\left(\frac{1}{n(n-1)}\right) = \sum_{\ell\ell'} \lambda_\ell \lambda_{\ell'} \left( \underbrace{4\bar{\pi}^2 \left(\frac{1}{2}(x_{\ell\ell'}^2 - x_\ell^2 x_{\ell'}^2) - (x_{\ell\ell'} - x_\ell x_{\ell'})x_\ell x_{\ell'}\right)}_{\text{var(III)}} \right.
$$

$$
\left. + \underbrace{4\pi^2 \left(\frac{1}{2}(y_{\ell\ell'}^2 - y_\ell^2 y_{\ell'}^2) - (y_{\ell\ell'} - y_\ell y_{\ell'})y_\ell y_{\ell'}\right)}_{\text{var(IV)}} \right)
$$

$$
\lesssim C^2 \|\lambda\|_2^2
$$

$$
\text{Coef}\left(\frac{1}{nm}\right) = \sum_{\ell\ell'} \lambda_\ell \lambda_{\ell'} \left( \underbrace{-(z_{\ell\ell'} - z_\ell z_{\ell'})x_\ell x_{\ell'} - (x_{\ell\ell'} - x_\ell x_{\ell'})z_\ell z_{\ell'} + (x_{\ell\ell'}z_{\ell\ell'} - x_\ell x_{\ell'}z_\ell z_{\ell'})}_{\text{var(I)}} \right.
$$

$$
\left. - \underbrace{(z_{\ell\ell'} - z_\ell z_{\ell'})y_\ell y_{\ell'} - (y_{\ell\ell'} - y_\ell y_{\ell'})z_\ell z_{\ell'} + (y_{\ell\ell'}z_{\ell\ell'} - y_\ell y_{\ell'}z_\ell z_{\ell'})}_{\text{var(I)}} \right)
$$

$$
\lesssim C^2 \|\lambda\|_2^2
$$

$$
\text{Coef}\left(\frac{1}{n^2}\right) = \sum_{\ell\ell'} \lambda_\ell \lambda_{\ell'} \left( \underbrace{(\bar{\pi} - \pi)\left(-(y_{\ell\ell'} - y_\ell y_{\ell'})x_\ell x_{\ell'} - (x_{\ell\ell'} - x_\ell x_{\ell'})y_\ell y_{\ell'} + (x_{\ell\ell'}y_{\ell\ell'} - x_\ell x_{\ell'}y_\ell y_{\ell'})\right)}_{\text{var(V)}} \right)
$$

$$
\lesssim C^2 \|\lambda\|_2^2.
$$

Summarizing, we've found that

$$
\text{var}(T(X,Y,Z) - \gamma(X,Y,\pi)) \lesssim \left(\frac{1}{n} + \frac{1}{m}\right) C\|\lambda\|_\infty (1 + R^2)\, \text{MMD}^2(P_X, P_Y)
$$
$$
+ \left(\frac{1}{n^2} + \frac{1}{nm}\right) C^2 \|\lambda\|_2^2. \tag{22}
$$

Using that $\langle \theta_r, \theta_{P_Y} - \theta_{P_X} \rangle_{\mathcal{H}_K} = 0$, we compute the expectation to be

$$
\mathbb{E}\left[-T(X,Y,Z) + \gamma(X,Y,\pi)\right] = (\pi - \nu)\, \text{MMD}^2(P_X, P_Y).
$$

Taking $\pi \triangleq \delta/2$ and applying Chebyshev's inequality shows that there exists a universal constant $c > 0$, such that the testing problem is possible at constant error probability (say $\alpha = 5\%$), provided that the sample sizes $m, n$ satisfy the following inequalities:

$$
\min\{m, n\} \geq c\frac{C\|\lambda\|_\infty(1 + R^2)}{\delta^2 \epsilon^2}
$$
$$
\min\{n, \sqrt{nm}\} \geq c\frac{C\|\lambda\|_2}{\delta \epsilon^2}.
$$

By repeated sample splitting and majority voting (see Appendix C), we can boost the success probability of this test to the desired level $1 - \alpha$ by incurring a multiplicative $\Theta(\log(1/\alpha))$ factor on the sample sizes $n, m$, which yields the desired result.

## E    Proof of Theorem 3.2

### E.1    Information theoretic tools

Our lower bounds rely on the method of two fuzzy hypotheses [50]. Given a measurable space $\mathcal{S}$, let $\mathcal{M}(\mathcal{S})$ denote the set of all probability measures on $\mathcal{S}$. We call subsets $H \subseteq \mathcal{M}(\mathcal{S})$ hypotheses. The following is the main technical result that our proofs rely on.

**Lemma E.1.** *Take hypotheses $H_0, H_1 \subseteq \mathcal{M}(\mathcal{S})$ and $P_0, P_1 \in \mathcal{M}(\mathcal{S})$ random with $\mathbb{P}(P_i \in H_i) = 1$. Then*

$$\inf_{\psi} \max_{i=0,1} \sup_{P \in H_i} P(\psi \neq i) \geq \frac{1}{2} \left(1 - \mathrm{TV}(\mathbb{E}P_0, \mathbb{E}P_1)\right),$$

*where the infimum is over all tests $\psi : \mathcal{X} \to \{0, 1\}$.*

*Proof.* For any $\psi$

$$\max_{i=0,1} \sup_{\mathbb{P}_i \in H_i} \mathbb{P}_i(\psi \neq i) \geq \frac{1}{2} \sup_{\mathbb{P}_i \in H_i} \left(\mathbb{P}_0(\psi = 1) + \mathbb{P}_1(\psi = 0)\right)$$

$$\geq \frac{1}{2} \mathbb{E}\left[P_0(\psi = 1) + P_1(\psi = 0)\right].$$

Optimizing over $\psi$ we get that the RHS above is equal to $\frac{1}{2}(1 - \mathrm{TV}(\mathbb{E}\, P_0, \mathbb{E}\, P_1))$ as required. $\qquad\square$

Therefore, to prove a lower bound on the minimax sample complexity of testing with total error probability $\alpha$, we just need to construct two random measures $P_i \in H_i$ such that $1 - \mathrm{TV}(\mathbb{E}\, P_0, \mathbb{E}\, P_1) = \Omega(\alpha)$. In our proofs we also use the following standard results on $f$-divergences.

**Lemma E.2** ([43, Section 7]). *For any probability distributions $P, Q$ the inequalities*

$$1 - \mathrm{TV}(P, Q) \geq \frac{1}{2} \exp(-\mathrm{KL}(P \| Q)) \geq \frac{1}{2} \frac{1}{1 + \chi^2(P \| Q)}$$

*hold.*

**Lemma E.3** (Chain rule for $\chi^2$-divergence). *Let $P_{X,Y}, Q_{X,Y}$ be probability measures such that the marginals on $X$ are equal ($P_X = Q_X$). Then*

$$\chi^2(P_{X,Y} \| Q_{X,Y}) = \chi^2(P_{Y|X} \| Q_{Y|X} | P_X).$$

*Proof.* Let $P_{X,Y}, Q_{X,Y}$ have densities $p, q$ with respect to some $\mu$. Then, by some abuse of notation, we have

$$\chi^2(P_{X,Y} \| Q_{X,Y}) = -1 + \int \frac{p(x,y)^2}{q(x,y)} \mathrm{d}\mu(x,y)$$

$$= -1 + \int \frac{p(y|x)^2 p(x)}{q(y|x)} \mathrm{d}\mu(x,y)$$

$$= \int p(x) \int \left(\frac{p(y|x)^2}{q(y|x)} - 1\right) \mathrm{d}\mu(y,x)$$

$$= \chi^2(P_{Y|X} \| Q_{Y|X} | P_X).$$

$\qquad\square$

## E.2  Constructing hard instances

Recall that in the statement of Theorem 3.2, we assume that $\mu(\mathcal{X}) = 1$, $\sup_{x \in \mathcal{X}} K(x, x) \leq 1$ and $\int K(x, y)\mu(\mathrm{d}x) \equiv \lambda_1$. Let $f_0 \equiv 1$ and for each $\eta \in \{\pm 1\}^{\mathbb{N}}$ define

$$f_\eta = 1 + \epsilon \underbrace{\sum_{j \geq 2} \rho_j \eta_j e_j}_{\triangleq g_\eta} \tag{23}$$

where $\{\rho_j\}_{j \geq 2}$ is chosen as $\rho_j = \mathbb{1}\{2 \leq j \leq J\}\sqrt{\lambda_j}/\|\lambda\|_{2,J}$, where we define $\|\lambda\|_{2,J} = \sqrt{\sum_{2 \leq j \leq J} \lambda_j^2}$ for some $J \geq 2$. Notice that $\int f_\eta(x)\mu(\mathrm{d}x) = \mu(\mathcal{X}) = 1$ due to orthogonality of the eigenfunctions. Assume from here on that $J$ is chosen so that for all $\eta$ we have $f_\eta(x) \geq 1/2$ for all $x \in \mathcal{X}$. This makes $f_\eta$ into a valid probability density with respect to the base measure $\mu$. Before continuing, we prove the following Lemma, which gives a lower bound on the maximal $J$ for which $f_\eta \geq 1/2$ for all $\eta$.

**Lemma E.4.** $J \leq J_\epsilon^\star$ holds provided $2\epsilon\sqrt{J-1} \leq \|\lambda\|_{2J}$.

*Proof of Lemma E.4.* Notice that

$$\|e_j\|_\infty = \sup_{x \in \mathcal{X}} \langle K(x, \cdot), e_j \rangle_\mathcal{H} \leq \sup_{x \in \mathcal{X}} \|K(x, \cdot)\|_\mathcal{H} \|e_j\|_\mathcal{H} \leq \frac{1}{\sqrt{\lambda_j}}, \tag{24}$$

where we use $\|K(x, \cdot)\|_\mathcal{H} = \sqrt{K(x, x)}$. We have

$$\|g_\eta\|_\infty = \epsilon\|\sum_{j \geq 2} \rho_j \eta_j e_j\|_\infty = \epsilon \sup_{x \in \mathcal{X}} \langle K(x, \cdot), \sum_{j \geq 2} \rho_j \eta_j e_j \rangle_\mathcal{H}$$

$$\leq \epsilon\|\sum_{j \geq 2} \rho_j \eta_j e_j\|_\mathcal{H} = \epsilon \sqrt{\sum_{j \geq 2} \rho_j^2 / \lambda_j} = \frac{\epsilon\sqrt{J-1}}{\|\lambda\|_{2,J}},$$

and the result follows. $\square$

Note that Lemma E.4 immediately gives us a proof of Corollary 3.5.

*Proof of Corollary 3.5.* Suppose that $J$ is such that $\sum_{j=2}^J \lambda_j^2 \geq c^2\|\lambda\|_2^2$. Then, by Lemma E.4, if $\epsilon \leq \|\lambda\|_{2J}/(2\sqrt{J-1})$ then $J \leq J_\epsilon^\star$. By assumption, this is implied by the inequality $\epsilon \leq c\|\lambda\|_2/(2\sqrt{J-1})$, and the result follows. $\square$

Continuing with our proof, note that by construction we have

$$\mathrm{MMD}^2(f_0, f_\eta) = \sum_{j \geq 2} \lambda_j \rho_j^2 = \epsilon^2, \quad \forall \eta \in \{\pm 1\}^\mathbb{N}. \tag{25}$$

### E.2.1 Lower Bound on $m$

Again, we apply Lemma E.1 with the new (deterministic) construction

$$P_0 = f_0^{\otimes n} \otimes (1+\epsilon e_2/\sqrt{\lambda_2})^n \otimes (1+\delta\epsilon e_2/\sqrt{\lambda_2})^{\otimes m}, \qquad P_1 = f_0^{\otimes n} \otimes (1+\epsilon e_2/\sqrt{\lambda_2})^n \otimes f_0^{\otimes m}, \tag{26}$$

where we write $f_\mathbb{1} = f_{(1,1,\dots)}$ and similarly for $g_\mathbb{1}$. By the data-processing inequality for $\chi^2$-divergence (also by Lemma E.3), we may drop the first $2n$ coordinates and obtain

$$\chi^2(\mathbb{E} P_0, \mathbb{E} P_1) = \chi^2((1+\delta\epsilon e_2/\sqrt{\lambda_2})^{\otimes m} \| f_0^{\otimes m})$$
$$= (1 + \delta^2\epsilon^2/\lambda_2)^m - 1$$
$$\leq \exp\left(\frac{\delta^2\epsilon^2 m}{\lambda_2}\right) - 1.$$

By Lemma E.2 we

$$1 - \mathrm{TV}(\mathbb{E} P_0, \mathbb{E} P_1) \gtrsim \frac{1}{\chi^2(\mathbb{E} P_0, \mathbb{E} P_1) - 1} \geq \exp(-\delta^2\epsilon^2 m) \stackrel{!}{=} \Omega(\alpha).$$

The lower bound $m \gtrsim \lambda_2 \log(1/\alpha)/(\delta\epsilon)^2$ now follows readily.

### E.2.2 Lower Bound on $n$

Once again, we apply Lemma E.1 to the new construction

$$P_0 = f_0^{\otimes n} \otimes f_\eta^{\otimes n} \otimes f_0^{\otimes m}, \qquad P_1 = f_\eta^{\otimes n} \otimes f_0^{\otimes n} \otimes f_0^{\otimes m}, \tag{27}$$

where we put a uniform prior on $\eta \in \{\pm 1\}^\mathbb{N}$ as before. Using the subadditivity of total variation under products, we compute

$$\mathrm{TV}(\mathbb{E} P_0, \mathbb{E} P_1) = \mathrm{TV}(f_0^{\otimes n} \otimes \mathbb{E} f_\eta^{\otimes n}, \mathbb{E}[f_\eta^{\otimes n}] \otimes f_0^{\otimes n})$$
$$\leq 2 \mathrm{TV}(\mathbb{E} f_\eta^{\otimes n}, f_0^{\otimes n}).$$

Just as in Appendix E.2.3 we upper bound by the $\chi^2$-divergence to get

$$\chi^2(\mathbb{E}\, f_\eta^{\otimes n} \| f_0^{\otimes n}) = -1 + \mathbb{E}_{\eta\eta'} \int \prod_{i=1}^n (f_\eta(x_i) f_{\eta'}(x_i)) \mu(\mathrm{d}x_1) \dots \mu(\mathrm{d}x_n)$$

$$\leq -1 + \mathbb{E}\exp(n\epsilon^2 \sum_{j \geq 2} \rho_j^2 \eta_j \eta_j')$$

$$= -1 + \prod_{j \geq 2} \cosh(n\epsilon^2 \rho_j^2)$$

$$\leq -1 + \exp(n^2 \epsilon^4 \sum_{j \geq 2} \rho_j^4)$$

$$= -1 + \exp(n^2 \epsilon^4 / \|\lambda\|_{2,J}^2).$$

Again, by Lemma E.2 we obtain

$$1 - \mathrm{TV}(\mathbb{E}\, P_0, \mathbb{E}\, P_1) \gtrsim \frac{1}{\chi^2(\mathbb{E}\, P_0 \| \mathbb{E}\, P_1) - 1} \geq \exp(-n^2 \epsilon^4 / \|\lambda\|_{2,J}^2) \overset{!}{=} \Omega(\alpha).$$

The lower bound $n \gtrsim \sqrt{\log(1/\alpha)} \|\lambda\|_{2,J} / \epsilon^2$ now follows readily.

### E.2.3   Lower Bound on $m \cdot n$

We take a uniform prior on $\eta$ and consider the random measures

$$P_0 = f_0^{\otimes n} \otimes f_\eta^{\otimes n} \otimes ((1-\delta)f_0 + \delta f_\eta)^{\otimes m} \qquad \text{and} \qquad P_1 = f_0^{\otimes n} \otimes f_\eta^{\otimes n} \otimes f_0^{\otimes m}. \tag{28}$$

Our goal is to apply Lemma E.1 to $P_0, P_1$. Notice that $(1-\delta)f_0 + \delta f_\eta = 1 + \delta\epsilon g_\eta$. Let us write $X, Y, Z$ for the marginals first $n$, second $n$ and last $m$ coordinates of $P_0$ and $P_1$. By the data processing inequality and the chain rule Lemma E.3 we have

$$\chi^2(\mathbb{E}\, P_0 \| \mathbb{E}\, P_1) = \chi^2((\mathbb{E}\, P_0)_{Y,Z} \| (\mathbb{E}\, P_1)_{Y,Z})$$

$$= \chi^2((\mathbb{E}\, P_0)_{Z|Y} \| (\mathbb{E}\, P_1)_{Z|Y} | (\mathbb{E}\, P_0)_Y)$$

$$= \mathbb{E}\, \chi^2 \left( \mathbb{E}\left[ (1 + \delta\epsilon g_\eta)^{\otimes m} \big| Y \right] \| f_0^{\otimes m} \right) =: (\dagger).$$

Notice that the expectation inside the $\chi^2$-divergence is with respect to $\eta$ given the variables $Y$, or in other words, over the posterior of $\eta$ with uniform prior given $n$ observations from the density $1 + \epsilon g_\eta = f_\eta$. The outer expectation is over $Y$. Given $Y$, let $\eta$ and $\eta'$ be i.i.d. from said posterior. We get the bound

$$(\dagger) + 1 \leq \mathbb{E} \int \prod_{i=1}^m (1 + \delta\epsilon g_\eta(x_i))(1 + \delta\epsilon g_{\eta'}(x_i)) \mu(\mathrm{d}x_i)$$

$$= \mathbb{E}(1 + \delta^2\epsilon^2 \sum_{j \geq 2} \rho_j^2 \eta_j \eta_j')^m$$

$$\leq \mathbb{E}\exp(\delta^2 \epsilon^2 m \sum_{j \geq 2} \rho_j^2 \eta_j \eta_j').$$

Define the collections of variables $\eta_{-j} = \{\eta_j\}_{j \geq 2} \setminus \{\eta_j\}$ and $\eta_{-j}'$ similarly. We shall prove the following claim:

$$\mathbb{E}\left[ \exp(\delta^2 \epsilon^2 m \rho_j^2 \eta_j \eta_j') \big| \eta_{-j} \eta_{-j}' \right] \leq \exp(c\delta^2 \epsilon^4 (\delta^2 m^2 + mn) \rho_j^4) \tag{29}$$

for some universal constant $c > 0$. Assuming that (29) holds, by induction we can show that

$$(\dagger) + 1 \leq \exp(c\delta^2 (\delta^2 m^2 + mn)\epsilon^4 \sum_{j \geq 2} \rho_j^4)$$

$$= \exp(c\delta^2 (\delta^2 m^2 + mn)\epsilon^4 / \|\lambda\|_{2,J}^2).$$

Thus, if $mn + \delta^2 m^2 = o\left( \|\lambda\|_{2,J}^2 / (\delta^2 \epsilon^4) \right)$ then testing is impossible.

We now prove (29). Since the variable $\eta_j'\eta_j'$ is either $1$ or $-1$, we have

$$\mathbb{E}\left[\exp(\delta^2\epsilon^2 m\rho_j^2\eta_j\eta_j')\big|\,\eta_{-j}\eta_{-j}'\right] = (e^{\delta^2\epsilon^2 m\rho_j^2} - e^{-\delta^2\epsilon^2 m\rho_j^2})\cdot\mathbb{P}(\eta_j\eta_j' = 1|\eta_{-j}\eta_{-j}') + e^{-\delta^2\epsilon^2 m\rho_j^2}.$$

Let us write $\eta_{\pm 1,j}$ for the vector of signs equal to $\eta$ but whose $j$'th coordinate is $\pm 1$ respectively. Looking at the probability above, and using the independence of $\eta, \eta'$ given $Y$, we have

$$\mathbb{P}(\eta_j\eta_j' = 1|Y,\eta_{-j},\eta_{-j}') = \mathbb{P}(\eta_j = 1|Y,\eta_{-j})^2 + \mathbb{P}(\eta_j = -1|Y,\eta_{-j})^2$$
$$= \frac{1}{4}\frac{(f_{\eta_{1j}}^{\otimes n}(Y))^2 + (f_{\eta_{-1j}}^{\otimes n}(Y))^2}{\left(\frac{1}{2}f_{\eta_{1j}}^{\otimes n}(Y) + \frac{1}{2}f_{\eta_{-1j}}^{\otimes n}(Y)\right)^2}.$$

Taking the expectation $\mathbb{E}[\cdot|\eta_{-j},\eta_{-j}']$ and using the HM-AM inequality $(\frac{1}{2}(x+y))^{-1} \leq \frac{1}{2}(\frac{1}{x} + \frac{1}{y})$ valid for all $x, y > 0$ gives

$$\mathbb{P}(\eta_j\eta_j' = 1|\eta_{-j},\eta_{-j}') = \frac{1}{4}\int\frac{(\prod_{i=1}^n f_{\eta_{1j}}(x_i))^2 + (\prod_{i=1}^n f_{\eta_{-1j}}(x_i))^2}{\frac{1}{2}\prod_{i=1}^n f_{\eta_{1j}}(x_i) + \frac{1}{2}\prod_{i=1}^n f_{\eta_{-1j}}(x_i)}\mu(\mathrm{d}x_1)\dots\mu(\mathrm{d}x_n)$$
$$\leq \frac{1}{4} + \frac{1}{8}\int\left(\frac{(\prod_{i=1}^n f_{\eta_{1j}}(x_i))^2}{\prod_{i=1}^n f_{\eta_{-1j}}(x_i)} + \frac{(\prod_{i=1}^n f_{\eta_{-1j}}(x_i))^2}{\prod_{i=1}^n f_{\eta_{1j}}(x_i)}\right)\mu(\mathrm{d}x_1)\dots\mu(\mathrm{d}x_n) = (\star).$$

Note that $f_{\eta_{1j}} = f_{\eta_{-1j}} + 2\epsilon\rho_j e_j$. Using the lower bound $f_{\eta_{\pm 1j}}(x) \geq \frac{1}{2}$ for all $x \in \mathcal{X}$ and the inequality $1 + x \leq \exp(x)$, we get

$$(\star) \leq \frac{1}{4} + \frac{1}{8}\left[\left(1 + \int\frac{4\epsilon^2\rho_j^2 e_j^2(x)}{f_{\eta_{-1j}}(x)}\mu(\mathrm{d}x)\right)^n + \left(1 + \int\frac{4\epsilon^2\rho_j^2 e_j^2(x)}{f_{\eta_{1j}}(x)}\mu(\mathrm{d}x)\right)^n\right]$$
$$\leq \frac{1}{4}(1 + e^{8\epsilon^2 n\rho_j^2}).$$

Recall that $(\star)$ is a probability so $(\star) \leq 1$, and we obtain

$$(\star) \leq \frac{1}{4}(1 + e^{8\epsilon^2 n\rho_j^2 \wedge \ln 3}).$$

Putting it together and applying Lemma E.5 we get

$$\text{LHS of (29)} \leq (e^{\delta^2\epsilon^2 m\rho_j^2} - e^{-\delta^2\epsilon^2 m\rho_j^2})\frac{1}{4}(1 + e^{8\epsilon^2 n\rho_j^2 \wedge \ln 3}) + e^{-\delta^2\epsilon^2 m\rho_j^2}$$
$$\leq e^{c\delta^2\epsilon^4\rho_j^4(\delta^2 m^2 + mn)}$$

for universal $c = 16 > 0$. Thus, by Lemma E.2 we obtain

$$1 - \text{TV}(\mathbb{E}\,P_0, \mathbb{E}\,P_1) \gtrsim \frac{1}{\chi^2(\mathbb{E}\,P_0, \mathbb{E}\,P_1) + 1} \geq \exp(-c\delta^2\epsilon^4(\delta^2 m^2 + mn)/\|\lambda\|_{2,J}^2) \overset{!}{=} \Omega(\alpha).$$

The necessity of

$$mn + \delta^2 m^2 \gtrsim \frac{\log(1/\alpha)\|\lambda\|_{2,J}^2}{\delta^2\epsilon^4}$$

follows immediately.[3]

**Lemma E.5.** *For $a, b \geq 0$, the following inequality holds:*

$$\frac{1}{4}(e^a - e^{-a})(1 + e^{b\wedge\ln 3}) + e^{-a} \leq e^{2(ab + a^2)}.$$

*Proof.* If $b \geq \ln 3$ or $a \geq 1$ we have:

$$\text{LHS} \leq \frac{1}{4}(e^a - e^{-a})(1 + e^{\ln 3}) + e^{-a} = e^a \leq e^{\frac{b}{\ln 3}a + a^2}.$$

If $b < \ln 3$ and $a < 1$, we have

$$e^b \leq 1 + \frac{2}{\ln 3}b \leq 1 + 2b, \quad \frac{e^a + e^{-a}}{2} \leq e^{a^2}, \quad \frac{e^a - e^{-a}}{2} \leq \frac{e - e^{-1}}{2}a \leq 2a,$$

---

[3]We have $mn + m^2 \leq (\sqrt{mn} + m)^2 \leq 2(mn + m^2)$, so $\sqrt{mn} + m \asymp \sqrt{mn + m^2}$.

and then

$$\frac{1}{4}(e^a - e^{-a})(1 + e^b) + e^{-a} = \frac{1}{2}(e^a + e^{-a}) + \frac{e^b - 1}{4}(e^a - e^{-a})$$

$$\leq e^{a^2} + 2ab$$

$$\leq e^{a^2}(1 + 2ab)$$

$$\leq e^{a^2 + 2ab}$$

The result follows from $\ln 3 > 1$. $\qquad\square$

## F    Proofs From Section 4

### F.1    Computing $\hat{\sigma}$

We follow the implementation of $\hat{\sigma}^2$ in [39]. Given $X_1, \ldots, X_{n_{\text{tr}}}^{\text{tr}}$ sampled from $P_X$ and $Y_1, \ldots, Y_{n_{\text{tr}}}^{\text{tr}}$ sampled from $P_Y$, denote

$$H_{ij} := K(X_i^{\text{tr}}, X_j^{\text{tr}}) + K(Y_i^{\text{tr}}, Y_j^{\text{tr}}) - K(X_i^{\text{tr}}, Y_j^{\text{tr}}) - K(Y_i^{\text{tr}}, X_j^{\text{tr}}), \quad i, j \in [n_{\text{tr}}]. \qquad (30)$$

Then $\hat{\sigma}^2$ is computed via

$$\hat{\sigma}^2(X^{n_{\text{tr}}}, Y^{n_{\text{tr}}}; K) = \frac{4}{n_{\text{tr}}^3} \sum_{i=1}^{n_{\text{tr}}} \left( \sum_{j=1}^{n_{\text{tr}}} H_{ij} \right)^2 - \frac{4}{n_{\text{tr}}^4} \left( \sum_{i=1}^{n_{\text{tr}}} \sum_{j=1}^{n_{\text{tr}}} H_{ij} \right)^2. \qquad (31)$$

Note that $\hat{\sigma}^2$ being non-negative follows from the AM-GM inequality.

### F.2    Heuristic Justification of the Objective (8)

As usual, let $X, Y, Z$ denotes samples of sizes $n, n, m$ from $P_X, P_Y, P_Z$ respectively. Let us give a heuristic justification for using the training objective defined in (8) for the purpose of obtaining a kernel for LFHT/mLFHT. Note that originally it was proposed as a training objective for kernels to be used in two sample testing. Recall that our test for LFHT can be written as

$$\Psi_{1/2}(X, Y, Z) = \mathbb{1}\left\{ T_{\text{LF}} \geq 0 \right\}$$

where

$$T_{\text{LF}} = \text{MMD}_u^2(\widehat{P}_Z, \widehat{P}_Y; K) - \text{MMD}_u^2(\widehat{P}_Z, \widehat{P}_X; K),$$

Heuristically, to maximize the power of (mLFHT), we would like to maximize the following population quantity

$$J_{\text{LF}} \triangleq \frac{\mathbb{E}_0[T_{\text{LF}}] - \mathbb{E}_1[T_{\text{LF}}]}{\sqrt{\text{var}_0(T_{\text{LF}})}}$$

where

$$\mathbb{E}_0[T_{\text{LF}}] = \mathbb{E}_{X,Y,Z}[T_{\text{LF}} | P_Z = P_X] = +\text{MMD}^2(P_X, P_Y; K),$$

$$\mathbb{E}_1[T_{\text{LF}}] = \mathbb{E}_{X,Y,Z}[T_{\text{LF}} | P_Z = P_Y] = -\text{MMD}^2(P_X, P_Y; K).$$

Let $T_{\text{TS}} = \text{MMD}_u(\hat{P}_X, \hat{P}_Y)$ be the usual statistic that is thresholded for two-sample testing. Then, a computation analogous to that in Section D.2 show (cf. (22)) that

$$\text{var}_0(T_{\text{LF}}) \approx \frac{A(K, P_X, P_Y)}{n} + \frac{A(K, P_X, P_Y)}{m} + \frac{B(K, P_X, P_Y)}{n^2} + \frac{B(K, P_X, P_Y)}{mn},$$

$$\text{var}_0(T_{\text{TS}}) \approx \frac{A(K, P_X, P_Y)}{n} + \frac{B(K, P_X, P_Y)}{n^2}$$

for some $A(K)$ and $B(K)$. Therefore, we have approximately

$$J_{\text{LF}} \approx \frac{2\,\text{MMD}^2(P_X, P_Y; K)}{\sqrt{1 + \frac{n}{m}}\sqrt{\text{var}_0(T_{\text{TS}})}} \approx 2\sqrt{\frac{m}{m+n}}\widehat{J}(X, Y; K)$$

which only differs from our optimization objective defined in (8) by a constant factor.

Second, notice that $\frac{\text{MMD}(P_X, P_Y; K)}{\sqrt{\text{var}(T_{\text{TS}})}}$ depends only on $P_X - P_Y$ and that $((1-\delta)P_X + \delta P_Y) - P_X \propto P_Y - P_X$, therefore it is sensible to use (8) as our training objective for is also sensible for (mLFHT), and we don't even need to observe the sample $Z$.

### F.3 Proof of Proposition 4.1

*Proof.* In this proof we regard $\mathcal{D} \triangleq (X^{\text{tr}}, X^{\text{ev}}, Y^{\text{tr}}, Y^{\text{ev}})$ and the parameters of the kernel $\omega$ as fixed. Recall that we are looking at the problem mLFHT with a misspecification parameter $R = 0$ (see Theorem 3.1). Given a test set $\{z_i\}_{i \in [m]}$, our test statistic is $T(\{z_i\}_{i \in [m]}) = \frac{1}{m} \sum_{i=1}^{m} f(z_i)$ where

$$f(z_i) = \frac{1}{n_{\text{ev}}} \sum_{j=1}^{n_{\text{ev}}} \left( K_\omega(z_i, Y_j^{\text{ev}}) - K_\omega(z_i, X_j^{\text{ev}}) \right).$$

In Phase 3 of Algorithm 1, we observe the value $\widehat{T} = T(Z) = \frac{1}{m} \sum_{i=1}^{m} f(Z_i)$ and reject the null hypothesis for large values of $\widehat{T}$. Thus, the $p$-value is defined as

$$p = p(Z, \mathcal{D}) \triangleq \mathbb{P}_{\widetilde{Z} \sim P_X^{\otimes m}}(T(\widetilde{Z}) > \widehat{T}).$$

Phase 2 of our Algorithm 1 produces random variables $T_1, \ldots, T_k$ that all have the distribution of $T(\{\widetilde{Z}_i\}_{i \in [m]})$, so that $\mathbb{1}\{T_r \geq \widehat{T}\}$ $(r = 1, \ldots, k)$ are unbiased estimates of the $p$-value. However, the $T_i$ are not independent, because they sample from the finite collection of calibration samples $X^{\text{cal}}$. However, as $n_{\text{cal}} \to \infty$ the covariances between $T_{r_1}, T_{r_2}$ for $r_1 \neq r_2$ tend to zero, and we obtain a consistent estimate of $p$. $\quad\square$

### F.4 Proof of Proposition 4.2

*Proof.* The test statistic $T(X, Y, Z)$ in (3) is given by

$$T(X, Y, Z) = \frac{1}{m} \sum_{i=1}^{m} f_K(Z_i)$$

where

$$f_K(z) = \theta_{\widehat{P}_Y}(z) - \theta_{\widehat{P}_X}(z).$$

This simplifies to (consider $K(x, y) = f(x)f(y)$)

$$f_K(z) = \left( \frac{1}{n} \sum_{j=1}^{n} f(Y_j) - \frac{1}{n} \sum_{j=1}^{n} f(X_j) \right) f(z) = C(X, Y)f(z).$$

where $C(X, Y)$ does not depend on $z$. Therefore, for any witness function $f$, we obtain the desired additive test. $\quad\square$

### F.5 Additive Test Statistics

In this section we prove accordingly that the test statistics of all of **MMD-M/G/O**, **SCHE**, **LBI**, **UME**, **RFM** are of the form $T_f(Z) = \frac{1}{m} \sum_{i=1}^{m} f(Z_i)$ (where $f$ might depends on $X, Y$). The test is to compare $T_f(Z)$ with some threshold $\gamma(X, Y)$.

Note that in the setting of Algorithm 1, the $X$ and $Y$ here correspond to $X^{\text{ev}}$ and $Y^{\text{ev}}$.

**MMD-M/G/O** As described in (3) we have

$$T_f(Z) = \frac{1}{m} \sum_{i=1}^{m} \left( \frac{1}{n} \sum_{j=1}^{n} (K(Z_i, Y_j) - K(Z_i, X_j)) \right).$$

**SCHE**  As described in Section 4.2 we have

$$T_f(Z) = \frac{1}{m} \sum_{i=1}^{m} \mathbb{1}\{\phi(Z_i) > t\}.$$

**LBI**  As described in Section 4.2 we have

$$T_f(Z) = \frac{1}{m} \sum_{i=1}^{m} \log\left(\frac{\phi(Z_i)}{1 - \phi(Z_i)}\right).$$

**UME**  As described in [31], the UME statistic evaluates the squared witness function at $J_q$ test locations $W = \{w_k\}_{k=1}^{J_q} \subset \mathcal{X}$. Formally for any two distributions $P, Q$ we define

$$U^2(P, Q) = \|\theta_Q - \theta_P\|_{L^2(W)}^2 = \frac{1}{J_q} \sum_{k=1}^{J_q} (\theta_Q(w_k) - \theta_P(w_k))^2.$$

However, we note a crucial difference that their result only considers the case of $n = m$, and their proposed estimator for $U^2(P_Z, P_X)$ can not be naturally extended to the case of $n \neq m$. Here we generalize it to $m \neq n$ where we (conveniently) use a biased estimate of their distance. Given samples $X, Y, Z$ and a set of witness locations $W$, the test statistic is a (biased yet) consistent estimator of $U^2(P_Z, P_Y) - U^2(P_Z, P_X)$. Let $\psi_W(z) = \frac{1}{\sqrt{J_q}}(K(z, w_1), \dots, K(z, w_{J_q})) \in \mathbb{R}^{|W|}$ be the "feature function," then:

$$\widehat{U}^2(Z, X) = \left\| \frac{1}{m} \sum_{i=1}^{m} \psi_W(Z_i) - \frac{1}{n} \sum_{j=1}^{n} \psi_W(X_i) \right\|_2^2$$

$$= \left\| \frac{1}{m} \sum_{i=1}^{m} \psi_W(Z_i) \right\|_2^2 + \left\| \frac{1}{n} \sum_{j=1}^{n} \psi_W(X_i) \right\|_2^2 - \frac{2}{mn} \sum_{1 \leq i \leq m, 1 \leq j \leq n} \langle \psi_W(Z_i), \psi_W(X_j) \rangle$$

Here $\langle \cdot, \cdot \rangle$ denotes the usual inner product. Therefore, the difference between distances is

$$\widehat{U}^2(Z, Y) - \widehat{U}^2(Z, X) = \frac{1}{m} \sum_{i=1}^{m} \left\langle \psi_W(Z_i), \frac{2}{n} \sum_{j=1}^{n} (\psi_W(X_j) - \psi_W(Y_j)) \right\rangle + F(X, Y)$$

where $F$ is sum function based only on $X, Y$. This is clearly an additive statistic for $Z$.

**RFM**  Algorithm 1 in [44] describes a method for learning a kernel from data given a binary classification task. For convenience lets concatenate the data to $X^{\text{RFM}} = (X, Y) \in \mathbb{R}^{2n \times d}$ and labels $y^{\text{RFM}} = (\vec{0}_n, \vec{1}_n) \in \mathbb{R}^{1 \times 2n}$. Given a learned kernel $K$, we write the Gram matrix as $(K(X^{\text{RFM}}, X^{\text{RFM}}))_{i,j} = K(X_i^{\text{RFM}}, X_j^{\text{RFM}})$ $(1 \leq i, j \leq 2n)$. Let $K(X^{\text{RFM}}, z)$ be a column vector with components $K(X_i^{\text{RFM}}, z)$ $(1 \leq i \leq 2n)$. The classifier is then defined as

$$f^{\text{RFM}}(z) = y^{\text{RFM}} \cdot K(X^{\text{RFM}}, X^{\text{RFM}})^{-1} \cdot K(X^{\text{RFM}}, z). \tag{32}$$

Though in [44] the kernel learned from RFM is used to construct a classifier as in Equation (32), since RFM is a feature learning method, we also apply the RFM kernel to our MMD test, namely

$$f^{\text{RFM to MMD}}(z) = \frac{1}{n} \sum_{j=1}^{n} (K(z, Y_j) - K(z, X_j)).$$

# G   Application: Diffusion Models vs CIFAR

We defer a more fine-grained detail to our code submission, which includes executable programs (with PyTorch) once the data-generating script from DDPM has been run (see README in the ./codes/CIFAR folder).

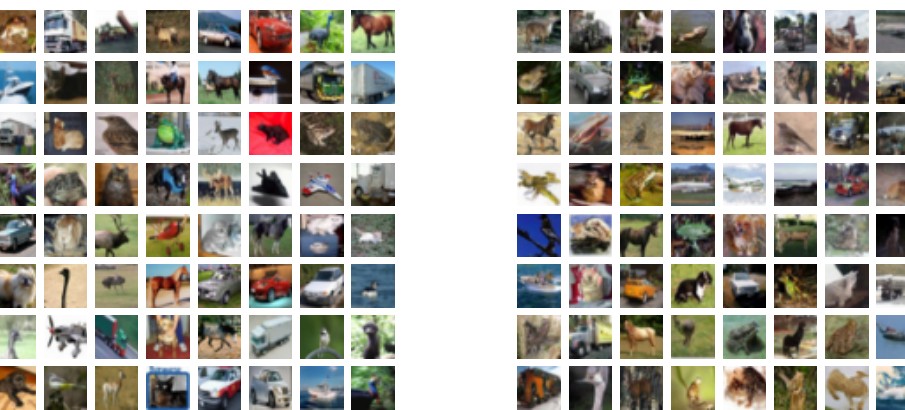

Figure 5: Data visualization for CIFAR-10 (left) vs DDPM diffusion generated images (right)

## G.1 Dataset Details

We use the CIFAR-10 dataset available online at https://www.cs.toronto.edu/~kriz/cifar.html, which contains 50000 colored images of size $32 \times 32$ with 10 classes. For the diffusion generated images, we use the SOTA Hugging Face model (DDPM) that can be found at https://huggingface.co/google/ddpm-CIFAR-10-32. We generated 10000 artificial images for our experiments. The code can be found at our code supplements.

For dataset balancing, we randomly shuffled the CIFAR-10 dataset and used 10000 images as data in our code. Most of our experiments are conducted with the null $P_X$ as CIFAR images, and the alternate as $P_Y = \frac{2}{3} \cdot \text{CIFAR} + \frac{1}{3} \cdot \text{DDPM}$. To this end, we matched 20000 images from CIFAR to belong to the alternate hypothesis, and the remaining 30000 images to stay in the null hypothesis. For the alternate dataset, we simply sample without replacement from the $20000 + 10000$ mixture. This sampled distribution is *almost* the same as mixing (so long as the sample bank is large enough compared to the acquired data, so that each item in the alternate has close to $1/3$ probability of being in DDPM, which is indeed the case).

## G.2 Experiment Setup and Benchmarks

We use a standard deep Conv-net [40], which has been employed for SOTA GAN discriminator tasks in similar settings. It has four convolutional layers and one fully connected layer outputting the feature space of size $(300, 1)$. For SCHE and LBI, we simply added a linear layer of $(300, 2)$ after applying ReLU to the 300-dimensional layer and used the cross-entropy loss to train the network. Note that this is equivalent to first fixing the feature space and then performing logistic regression to the feature space. For kernels, we add extra trainable parameters after the 300-d feature output.

For the MMD-based tests, we simply train the kernel on the neural net and evaluate our objective. For UME, we used a slightly generalized version of the original statistic in [31] which allows for comparison on randomly selected witness locations in the null hypothesis with $m \neq n$ (see Appendix F.5). The kernel is trained using our heuristic (see (8) and Appendix F.2), with MMD replaced by UME. The formula for UME variance can be found in [31]. For RFM, we use Algorithm 1 in [44] to learn a kernel on (stochastic batched) samples, and then use our MMD test on the trained kernel.

We use 80 training epochs for most of our code from the CNN architecture (for classifiers, this is well after interpolating the training data and roughly when validation loss stops decreasing), and a batch size of 32 which has a slight empirical benefit compared to larger batch sizes. The learning rates are tuned separately in MMD methods for optimality, whereas for classifiers they follow the discriminator's original setting from [40]. In Phase 2 of Algorithm 1, we choose $k = 1000$ for the

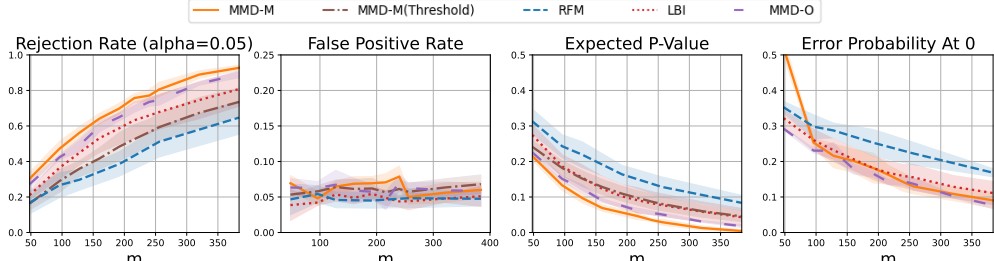

Figure 6: Relevant plots following the setting in Figure 3 (in the main text) of fixing $n_{\text{tr}} = 1920$ and varying sample size $m$ in the x-axis for the comparison with missing benchmarks. Errorbars are projected showing standard deviation across 10 runs. We replaced part (d) in Figure 3 (in the main text) to a sanity check in our FPR when thresholded at $\alpha = 0.05$.

desired precision while not compromising runtime. For each task, we run 10 independent models and report their performances as the mean and standard deviation of those 10 runs as estimates. We refer to a full set of hyper-parameters in our code implementation.

Our code is implemented in Python 3.7 (PyTorch 1.1) and was ran on an NVIDIA RTX 3080 GPU equipped with a standard torch library and dataset extensions. Our code setup for feature extraction is similar to that of [39]. For benchmark implementations, our code follows from the original code templated provided by the cited papers.

### G.3 Sample Allocation

We make a comment on why (4) is *different* from just thresholding $\widehat{\text{MMD}}^2(Z, Y^{\text{tr}}) - \widehat{\text{MMD}}^2(Z, X^{\text{tr}})$ at 0, which was what we did in part (c) of Figure 3 (and hence the difference along the curve of MMD-M vs Figure 1.2). Our theory assumes that the samples are i.i.d. conditioned on the kernel being chosen already. However, in the experiments, the kernel is dependent on the training data. Therefore, to evaluate the MMD estimate (between experimentations), one needs extra data that does not intersect with training.

In fact, it can be experimentally shown by comparing Figure 1.2 and Figure 2(c) that doing so (while reducing the sample complexity on $n_{\text{ev}}$) hurts performance. Indeed, we found out that when $X^{\text{ev}}, Y^{\text{ev}}$ are non-intersecting with training, performance is (almost) always better at a cost of hurting the overall sample complexity of $n$.

### G.4 Remarks on Results

Figure 6 lists all of our benchmarks in the setting of Figure 3 (in the main text) on missing benchmarks, where the last figure is replaced by the false positive rate at thresholding at $\alpha = 0.05$ to verify our results. As mentioned in the main text, our MMD-M method consistently outperforms other benchmarks on both the expected $p$-value (of alternate) and rejection rate at $\alpha = 0.05$, while all of our tests observe a empirical false positive rate close to $\alpha = 0.05\%$ (Part (b)), showing the consistency of methods.

## H Application: Higgs-Boson Detection

### H.1 Dataset Details

We use the Higgs dataset available online at http://archive.ics.uci.edu/ml/datasets/HIGGS, produced using Monte Carlo simulations [5]. The dataset is nearly balanced, containing $5,829,122$ signal instances and $5,170,877$ background instances. Each instance is a 28-dimensional vector, consisting of 28 features. The first 21 features are kinematic properties measured by the detectors in the accelerator, such as momentum and energy. The last 7 properties are *invariant masses*, derived from the first 21 features.

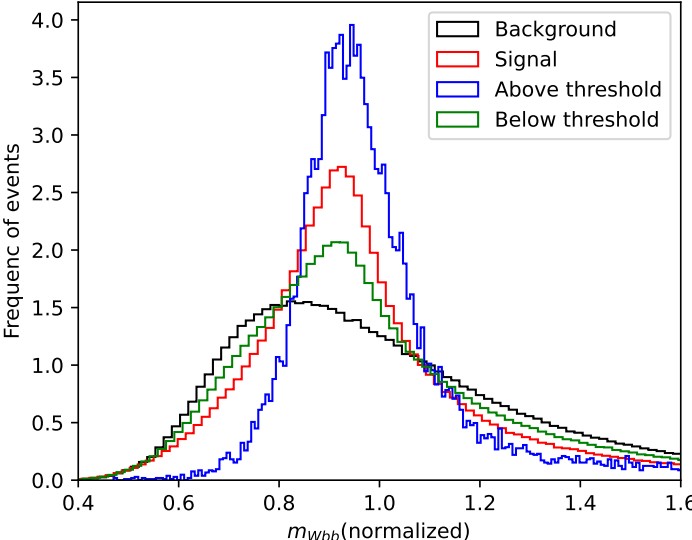

Figure 7: This figure visualizes the distribution of the 26th feature, the invariant mass $m_{Wbb}$. The red and black lines are the histograms of the original dataset. We employ MMD-M as a classifier, trained and evaluated using $n_{tr} = 1.3 \times 10^6$ and $n_{ev} = n_{opt} = 2 \times 10^4$ through Algorithm 3. The blue(green) line represents all instances $z$'s whose "witness scores" $f(z; X^{ev}, Y^{ev})$'s are larger(smaller) than $t_{opt}$.

## H.2 Experiment Setup and Training Models

The modified Algorithm 1 is shown in Algorithm 2 and Algorithm 3. Compared with Algorithm 2, we implement the thresholding trick (Section 4.3) in Algorithm 3.

### H.2.1 Configuration and Model Architecture

We implement all methods in Python 3.9 and PyTorch 1.13 and run them on an NVIDIA Quadro RTX 8000 GPU.

For all classifier-based methods in this study (SCHE and LBI), we adopt the same architecture as previously proposed in [5]. The classifiers are six-layer neural networks with 300 hidden units in each layer, all employing the tanh activation function. For SCHE, the output layer is a single sigmoid unit and we utilize the binary cross-entropy loss for training. For LBI, the output layer is a linear unit and we utilize the binary cross entropy loss combined with a logit function (which is more numerically stable than simply using a sigmoid layer followed by a cross entropy loss).

For all MMD-based methods (MMD-M, MMD-G, MMD-O, and UME), the networks $\varphi$ and $\varphi'$ are both six-layer neural networks with 300 ReLU units in each layer. The feature space, which is the output of the neural network $\varphi$, is set to be 100-dimensional. Here UME has the same kernel architecture as MMD-M, and the number of test locations is set to be $J_q = 4096$. For RFM, we adopt the same architecture as in [44], where the kernel is $K_M(x, y) = \exp(-\gamma(x - y)^T M(x - y))$ with a constant $\gamma$ and a learnable positive semi-definite matrix $M$. We set $\gamma \equiv 1$.

The neural networks are initialized using the default setting in PyTorch, and the bandwidths $\sigma, \sigma'$ are initialized using the *median heuristic* [22]. The parameter $\tau$ is initially set to 0.5. For UME, the witness locations $W$ are initially randomly sampled from the training set. For RFM, the initial $M$ equals the median bandwidth times an identity matrix.

### H.2.2 Training

The size of our training set, denoted as $n_{tr}$, varies from $1.0 \times 10^2$ to $1.6 \times 10^6$. For a given $n_{tr}$, we select the first $n_{tr}$ datapoints from each class of the Higgs dataset to form $X^{tr}$ and $Y^{tr}$, i.e., $|X^{tr}| = |Y^{tr}| = n_{tr}$. Subsequently, we randomly select $n_{validation} = \min(\sqrt{10 n_{tr}}, 0.1 n_{tr})$ points

from each of $X_{tr}, Y_{tr}$ to constitute the validation set, while the remainder of $X_{tr}, Y_{tr}$ are used for running gradient descent. The optimizer is set to be a minibatch SGD, with a batch size of 1024, a learning rate of 0.001, and a momentum of 0.99. Training is halted once the validation loss stops to decrease for 10 epochs, then we choose the checkpoint (saved for each epoch) with the smallest validation loss thus far as our trained model. Beyond the general setting above, in RFM a batch size of 1024 doesn't work well and instead we use a batch size of 20,000.

### H.3    Evaluating the Performance

#### H.3.1    Evaluating the p-Value with the Methodology of Algorithm 1

We call the "witness score" of an instance $z \in \mathcal{X}$ as

$$f(z; X^{ev}, Y^{ev}) = \frac{1}{n_{cal}} \sum_{i=1}^{n_{cal}} \left( k(z, Y_i^{ev}) - k(z, X_i^{ev}) \right). \tag{33}$$

For a vector of instances $Z = (Z_1, \ldots, Z_m)$, we write

$$f(Z; X^{ev}, Y^{ev}) = (f(Z_1; X^{ev}, Y^{ev}), \ldots, f(Z_m; X^{ev}, Y^{ev})).$$

The testing procedure is summarized in Phases 2, 3 and 4 in Algorithm 2 and Algorithm 3. In the Higgs experiment, we utilize the Gaussian approximation method to determine the p-values when the witness function $f$ is not thresholded, which allows us to reach very small p-values and errors under limited computational resource. In cases where the score function $f$ is thresholded by a value $t$, using the Binomial distribution as in Algorithm 3 is more precise and also fast enough.

Given a trained kernel $K$ trained on $X^{tr}$ and $Y^{tr}$, we set $X^{ev} = X^{tr}$ and $Y^{ev} = Y^{tr}$, and accordingly $n_{ev} = n_{tr}$. This results in a more efficient use of data (since we reuse $X^{tr}, Y^{tr}$ also as $X^{ev}, Y^{ev}$). Then, out of the untouched portion of the data, we randomly choose $n_{cal} = 20,000$ datapoints from both classes to populate $X^{cal}$ and $Y^{cal}$, i.e., $|X^{cal}| = |Y^{cal}| = n_{cal} = 20,000$. In addition to the general setting above, for RFM, we need to solve a $2n_{ev}$-dimensional linear equation during inference, which arises from the inverse matrix in Equation (32) (solving $K(X^{RFM}, X^{RFM})\boldsymbol{u} = (y^{RFM})^T$ for $\boldsymbol{u} \in \mathbb{R}^{2n_{ev}}$). So we set $n_{ev} = \min(n_{tr}, 10,000)$ that $X_{ev}, Y_{ev}$ are randomly sampled from the training set.

In order to compare different benchmarks, we evaluate the expected significance of discovery on a mixture of 1000 backgrounds and 100 signals. For each benchmark and each $n_{tr}$, we train 10 independent models. Then for each trained model we proceed through the Phases 2, 3 (and 4) in Algorithm 2 and Algorithm 3 by 10 times for 10 different $(X^{ev}, X^{cal}, X^{opt}, Y^{ev}, Y^{cal}, Y^{opt})$. The mean and standard deviation from these 100 runs are reported in Figure 8.

We also display in Figure 9 the trade-off b $(m, n_{ev})$ and $(m, n_{tr})$ to reach certain levels of significance of discovery in MMD-M. From the bottom left plot, we see that the (averaged) significance is not sensitive to $n_{ev}$ when $\lg n_{ev}$ is large. So taking $n_{ev} = 20,000$ is sufficient.

#### H.3.2    Evaluating the Error of the Test (4)

We set the parameters to be $\delta = 0.1$ and $\pi = \frac{1}{2}\delta$ in our experiments. As explained Appendix G.3, here we no longer take $X^{ev} = X^{tr}$. Empirically, taking $X^{ev} = X^{tr}$ yields a very bad threshold $\gamma(X^{ev}, Y^{ev}, \pi)$.[4] Instead, $X^{ev}$ is sampled from untouched datapoints other than $X^{tr}$, and the same applies for $Y$. We still take $n_{ev} = n_{tr}$ here, resulting in a total size of $n_{ev} + n_{tr} = 2n_{tr}$. Specifically, when $n_{ev} \geq 10,000$, computing a $n_{ev} \times n_{ev}$ Gram matrix becomes computationally expensive, so we adopt Monte Carlo method to compute $\gamma(X^{ev}, Y^{ev}, \pi)$, in which we subsample 10,000 points from $X^{ev}$ and $Y^{ev}$ to calculate $\gamma$ and repeat this process 100 times.

Again, we utilize the Gaussian approximation. Recall that the test is to compare $T = \frac{1}{m} \sum_{i=1}^{m} f(Z_i)$ with $\gamma$. The type 1 and type 2 error are estimated as $\text{CDF}_{\mathcal{N}(0,1)} \left( -\frac{\gamma(X^{ev}, Y^{ev}, \pi) - \mathbb{E}[f|H_0]}{\sqrt{\text{var}(f|H_0)/m}} \right)$ and

---

[4]If the kernel $K(\cdot, \cdot) = K_{X^{tr}, Y^{tr}}(\cdot, \cdot)$ is independent of $X^{ev}, Y^{ev}$, then we have $\gamma(X^{ev}, Y^{ev}, \delta/2) \approx \frac{1}{2} \left( \mathbb{E}_{Z \sim P_x}[T(X^{ev}, Y^{ev}, Z)] + \mathbb{E}_{Z \sim \delta P_Y + (1-\delta)P_X}[T(X^{ev}, Y^{ev}, Z)] \right)$. However this is no longer true if $(X^{tr}, Y^{tr})$ and $(X^{ev}, Y^{ev})$ intersect.

$\text{CDF}_{\mathcal{N}(0,1)} \left( -\frac{\mathbb{E}[f|H_1] - \gamma(X^{\text{ev}}, Y^{\text{ev}}, \pi)}{\sqrt{\text{var}(f|H_1)/m}} \right)$ for the witness function $f$, which can be estimated efficiently using the calibration samples $X^{\text{cal}}, Y^{\text{cal}}$.

We consider both the regimes of fixing kernels and varying kernels (training kernel based on $n$). The results are shown in the top plot in Figure 1.2 and the top plot in Figure 9. For each point on the plot, we train 30 independent models and test each model 10 times, and report the average of these 300 runs. In both plots, we observe the asymmetric $m$ vs $n$ trade-off.

---

**Algorithm 2** Estimate the significance of discovery of an input $Z_{\text{test}}$, using the original statistic

---

**Input:** $(X^{\text{tr}}, X^{\text{ev}}, X^{\text{cal}}), (Y^{\text{tr}}, Y^{\text{ev}}, Y^{\text{cal}})$; parametrized kernel $K_\omega$; input $Z_{\text{test}}$.
*# Phase 1: Kernel training on $X^{\text{tr}}$ and $Y^{\text{tr}}$*
$\omega \leftarrow \arg\max_\omega^{\text{optimizer}} \hat{J}(X^{\text{tr}}, Y^{\text{tr}}; K_w)$        *# maximize objective $\hat{J}(X^{\text{tr}}, Y^{\text{tr}}; K_\omega)$ as in (8)*
*# Phase 2: Distributional calibration of test statistic*
$\text{Scores}^{(0)} \leftarrow f(X^{\text{cal}}; X^{\text{ev}}, Y^{\text{ev}})$        *# $\text{Scores}^{(0)}$ has a length of $n_{\text{cal}}$*
$\text{Scores}^{(1)} \leftarrow f(Y^{\text{cal}}; X^{\text{ev}}, Y^{\text{ev}})$        *# $\text{Scores}^{(1)}$ has a length of $n_{\text{cal}}$*
$\theta_0 \leftarrow \text{mean}(\text{Scores}^{(0)})$        *# estimate $\mathbb{E}[f(Z)|Z \sim P_X]$*
$\theta_1 \leftarrow \text{mean}(\text{Scores}^{(1)})$        *# estimate $\mathbb{E}[f(Z)|Z \sim P_Y]$*
$\sigma_0 \leftarrow \text{std}(\text{Scores}^{(0)})$        *# estimate $\sqrt{\text{var}[f(Z)|Z \sim P_X]}$*
*# Phase 3: Inference with input $Z_{test}$*
$m \leftarrow \text{length}(Z_{\text{test}})$
$T \leftarrow T_f(Z_{\text{test}}; X^{\text{ev}}, Y^{\text{ev}}) = \text{mean}(f(Z_{\text{test}}; X^{\text{ev}}, Y^{\text{ev}}))$        *# compute test statistic*
$Z_{\text{discovery}} \leftarrow \frac{T - \theta_0}{\sigma_0/\sqrt{m}}$
**Output:** Estimated significance: $Z_{\text{discovery}}$

---

---

**Algorithm 3** Estimate the significance of discovery of an input $Z_{\text{test}}$, applying the thresholding trick

---

**Input:** $(X^{\text{tr}}, X^{\text{ev}}, X^{\text{cal}}, X^{\text{opt}}), (Y^{\text{tr}}, Y^{\text{ev}}, Y^{\text{cal}}, Y^{\text{opt}})$; parametrized kernel $K_\omega$; input $Z_{\text{test}}$.
*# Phase 1: Kernel training on $X^{\text{tr}}$ and $Y^{\text{tr}}$*
$\omega \leftarrow \arg\max_\omega^{\text{optimizer}} \hat{J}(X^{\text{tr}}, Y^{\text{tr}}; K_w)$        *# maximize objective $\hat{J}(X^{\text{tr}}, Y^{\text{tr}}; K_\omega)$ as in (8)*
*# Phase 2: Find the best threshold*
$\text{Scores}^{(0)} \leftarrow f(X^{\text{opt}}; X^{\text{ev}}, Y^{\text{ev}})$
$\text{Scores}^{(1)} \leftarrow f(Y^{\text{opt}}; X^{\text{ev}}, Y^{\text{ev}})$        *# witness function as in (33)*
**for** $i = 1, 2, ..., 2n_{\text{opt}}$ **do**
   $t = (\text{Scores}^{(0)} \cup \text{Scores}^{(1)})[i]$
   $\text{TP}, \text{TN} = \text{mean}(\text{Scores}^{(1)} > t), \text{mean}(\text{Scores}^{(0)} < t)$    *# true positive and true negative rate*
   $\text{power}_i = \frac{\text{TP} + \text{TN} - 1}{\sqrt{\text{TN}(1 - \text{TN})}}$        *# find $t$ to maximize the (estimated) p-value*
**end for**
$t_{\text{opt}} = (\text{Scores}^{(0)} \cup \text{Scores}^{(1)})[\arg\max_i \text{power}_i]$
*# Phase 3: Distributional calibration of test statistic (under null hypothesis)*
$\text{Scores}^{(0)} \leftarrow (f(X^{\text{cal}}; X^{\text{ev}}, Y^{\text{ev}}) > t)$        *# $\text{Scores}^{(0)} \in \{0, 1\}^{n_{\text{ev}}}$*
$\text{Scores}^{(1)} \leftarrow (f(Y^{\text{cal}}; X^{\text{ev}}, Y^{\text{ev}}) > t)$        *# $\text{Scores}^{(1)} \in \{0, 1\}^{n_{\text{ev}}}$*
$\theta_0 \leftarrow \text{mean}(\text{Scores}^{(0)})$        *# estimate $\mathbb{E}[f_t(Z)|Z \sim P_X] \in [0, 1]$*
$\theta_1 \leftarrow \text{mean}(\text{Scores}^{(1)})$        *# estimate $\mathbb{E}[f_t(Z)|Z \sim P_Y] \in [0, 1]$*
*# Phase 4: Inference with input $Z_{test}$*
$m \leftarrow \text{length}(Z_{\text{test}})$
$T \leftarrow T_f(Z_{\text{test}}; X^{\text{ev}}, Y^{\text{ev}}) = \text{mean}(f(Z_{\text{test}}; X^{\text{ev}}, Y^{\text{ev}}) > t)$        *# compute test statistic*
$Z_{\text{discovery}} \leftarrow \text{CDF}_{\mathcal{N}(0,1)}^{-1}(\text{CDF}_{\text{Bin}(m, \theta_0)}(T))$
**Output:** Estimated significance: $Z_{\text{discovery}}$

---

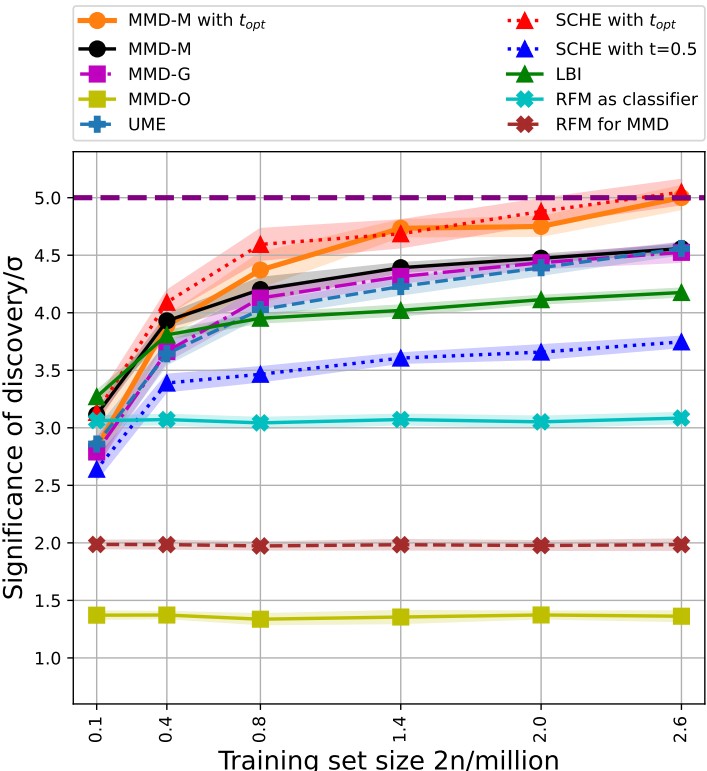

Figure 8: Complete image of Figure 1.2 in the main text. The mean and standard deviation are calculated based on 100 runs. See Appendix H for details.

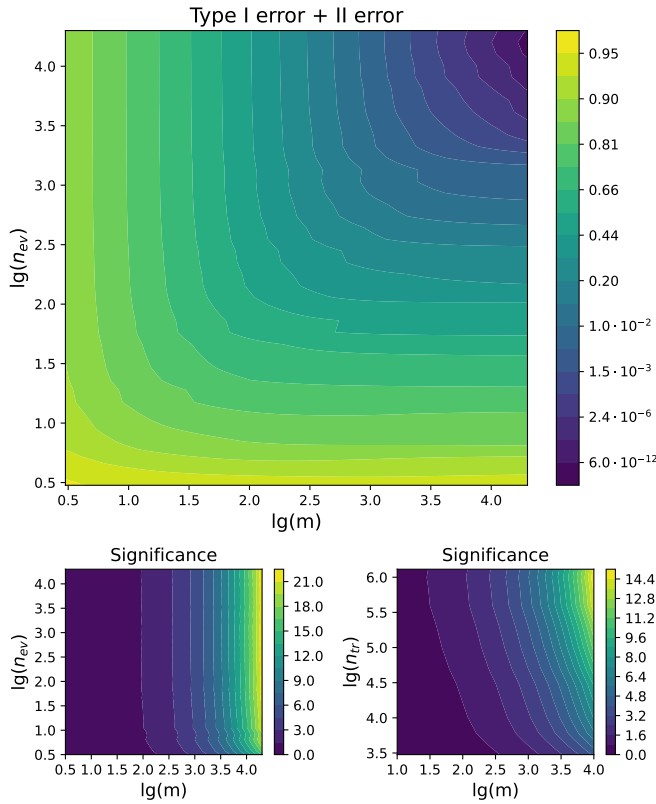

Figure 9: The top plot displays the $(m, n_{\text{ev}})$ trade-off to reach certain levels of total error using $n_{\text{tr}} = 1.3 \times 10^6$ in MMD-M. The bottom figures show the trade-off of $(m, n_{\text{ev}})$ and $(m, n_{\text{tr}})$ to reach certain level of significance of discovery in MMD-M. In the bottom left figure, we fix $n_{\text{tr}} = 1.3 \times 10^6$. In the bottom right figure, we fix $n_{\text{ev}} = 20,000$. See Appendix H for details.

## I  Limitations and Future Directions

Finally, we discuss several limitations of our work and raise open questions that we hope will be addressed in future works. From the theoretical side of our arguments, we point out several aspects. First, our upper bound (on the minimax sample complexity) Theorem 3.1 has a likely sub-optimal dependence on $\alpha, \delta$. Second, it might be possible to improve our lower bound to a more natural form by replacing $\|\lambda\|_{2, J_\epsilon^\star}$ by $\|\lambda\|_2$ and removing the constraint that the top eigenfunction has to be constant. Third, it remains open to extend our theory to include data-dependent $K$, as opposed to fixed $K$.

Empirically, our proposal Algorithm 1 can be inefficient in Phase 2 (prior works such as [39] have used permutation-based arguments for a more efficient estimate), which we adopted due to its simplicity and universality in all benchmarks. Moreover, one might hope that LFHT/mLFHT can be extended to more complex applications, such as text data or videos. Such questions are important to investigate as a future direction.

