**Lemma E.5.** *For $a, b \ge 0$, the following inequality holds:*

$$\frac{1}{4}(e^a - e^{-a})(1 + e^{b\wedge\ln 3}) + e^{-a} \le e^{2(ab+a^2)}.$$

*Proof.* If $b \ge \ln 3$ or $a \ge 1$ we have:

$$\text{LHS} \le \frac{1}{4}(e^a - e^{-a})(1 + e^{\ln 3}) + e^{-a} = e^a \le e^{\frac{b}{\ln 3}a + a^2}.$$

---

[3] We have $mn + m^2 \le (\sqrt{mn} + m)^2 \le 2(mn + m^2)$, so $\sqrt{mn} + m \asymp \sqrt{mn + m^2}$.

If $b < \ln 3$ and $a < 1$, we have

$$e^b \le 1 + \frac{2}{\ln 3} b \le 1 + 2b, \quad \frac{e^a + e^{-a}}{2} \le e^{a^2}, \quad \frac{e^a - e^{-a}}{2} \le \frac{e - e^{-1}}{2} a \le 2a,$$

and then

$$
\begin{aligned}
\frac{1}{4}(e^a - e^{-a})(1 + e^b) + e^{-a} &= \frac{1}{2}(e^a + e^{-a}) + \frac{e^b - 1}{4}(e^a - e^{-a}) \\
&\le e^{a^2} + 2ab \\
&\le e^{a^2}(1 + 2ab) \\
&\le e^{a^2 + 2ab}
\end{aligned}
$$

The result follows from $\ln 3 > 1$. $\qquad\square$

## F   Proofs From Section 4

### F.1   Heuristic Justification of the Objective (7)

As usual, let $X, Y, Z$ denotes samples of sizes $n, n, m$ from $P_X, P_Y, P_Z$ respectively. Let us give a heuristic justification for using the training objective defined in (7) for the purpose of obtaining a kernel for LFHT/mLFHT. Note that originally it was proposed as a training objective for kernels to be used in two sample testing. Recall that our test for LFHT can be written as

$$\Psi_{1/2}(X, Y, Z) = \mathbb{1}\left\{ T_{\mathsf{LF}} \ge 0 \right\}$$

where

$$T_{\mathsf{LF}} = \mathrm{MMD}_u^2(\widehat{P}_Z, \widehat{P}_Y; K) - \mathrm{MMD}_u^2(\widehat{P}_Z, \widehat{P}_X; K),$$

Heuristically, to maximize the power of (mLFHT), we would like to maximize the following population quantity

$$J_{\mathsf{LF}} \triangleq \frac{\mathbb{E}_0[T_{\mathsf{LF}}] - \mathbb{E}_1[T_{\mathsf{LF}}]}{\sqrt{\mathrm{var}_0(T_{\mathsf{LF}})}}$$

where

$$
\begin{aligned}
\mathbb{E}_0[T_{\mathsf{LF}}] &= \mathbb{E}_{X,Y,Z}[T_{\mathsf{LF}}|P_Z = P_X] = +\mathrm{MMD}^2(P_X, P_Y; K), \\
\mathbb{E}_1[T_{\mathsf{LF}}] &= \mathbb{E}_{X,Y,Z}[T_{\mathsf{LF}}|P_Z = P_Y] = -\mathrm{MMD}^2(P_X, P_Y; K).
\end{aligned}
$$

Let $T_{\mathsf{TS}} = \mathrm{MMD}_u(\hat{P}_X, \hat{P}_Y)$ be the usual statistic that is thresholded for two-sample testing. Then, a computation analogous to that in Section D.2 show (cf. (13)) that

$$
\begin{aligned}
\mathrm{var}_0(T_{\mathsf{LF}}) &\approx \frac{A(K)}{n} + \frac{A(K)}{m} + \frac{B(K)}{n^2} + \frac{B(K)}{mn}, \\
\mathrm{var}_0(T_{\mathsf{TS}}) &\approx \frac{A(K)}{n} + \frac{B(K)}{n^2}
\end{aligned}
$$

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

^{\text{ev}}, Y^{\text{ev}}) = \frac{1}{n_{\text{cal}}} \sum_{i=1}^{n_{\text{cal}}} \left( k(z, Y_i^{\text{ev}}) - k(z, X_i^{\text{ev}}) \right). \tag{22}$$

For a vector of instances $Z = (Z_1, \ldots, Z_m)$, we write

$$f(Z; X^{\text{ev}}, Y^{\text{ev}}) = (f(Z_1; X^{\text{ev}}, Y^{\text{ev}}), \ldots, f(Z_m; X^{\text{ev}}, Y^{\text{ev}})).$$

The testing procedure is summarized in Phases 2, 3 and 4 in Algorithm 2 and Algorithm 3. In the Higgs experiment, we utilize the Gaussian approximation method to determine the p-values when the witness function $f$ is not thresholded, which allows us to reach very small p-values and errors under limited computational resource. In cases where the score function $f$ is thresholded by a value $t$, using the Binomial distribution as in Algorithm 3 is more precise and also fast enough.

Given a trained kernel $K$ trained on $X^{\text{tr}}$ and $Y^{\text{tr}}$, we set $X^{\text{ev}} = X^{\text{tr}}$ and $Y^{\text{ev}} = Y^{\text{tr}}$, and accordingly $n_{\text{ev}} = n_{\text{tr}}$. This results in a more efficient use of data (since we reuse $X^{\text{tr}}, Y^{\text{tr}}$ also as $X^{\text{ev}}, Y^{\text{ev}}$). Then, out of the untouched portion of the data, we randomly choose $n_{\text{cal}} = 20,000$ datapoints from both classes to populate $X^{\text{cal}}$ and $Y^{\text{cal}}$, i.e., $|X^{\text{cal}}| = |Y^{\text{cal}}| = n_{\text{cal}} = 20,000$. In addition to the general setting above, for RFM, we need to solve a $2n_{\text{ev}}$-dimensional linear equation during inference, which arises from the inverse matrix in Equation (21) (solving $K(X^{\text{RFM}}, X^{\text{RFM}})\boldsymbol{u} = (y^{\text{RFM}})^T$ for $\boldsymbol{u} \in \mathbb{R}^{2n_{\text{ev}}}$). So we set $n_{\text{ev}} = \min(n_{\text{tr}}, 10,000)$ that $X_{\text{ev}}, Y_{\text{ev}}$ are randomly sampled from the training set.

In order to compare different benchmarks, we evaluate the expected significance of discovery on a mixture of 1000 backgrounds and 100 signals. For each benchmark and each $n_{\text{tr}}$, we train 10 independent models. Then for each trained model we proceed through the Phases 2, 3 (and 4) in Algorithm 2 and Algorithm 3 by 10 times for 10 different $(X^{\text{ev}}, X^{\text{cal}}, X^{\text{opt}}, Y^{\text{ev}}, Y^{\text{cal}}, Y^{\text{opt}})$. The mean and standard deviation from these 100 runs are reported in Figure 7.

We also display in Figure 8 the trade-off b $(m, n_{\text{ev}})$ and $(m, n_{\text{tr}})$ to reach certain levels of significance of discovery in MMD-M. From the bottom left plot, we see that the (averaged) significance is not sensitive to $n_{\text{ev}}$ when $\lg n_{\text{ev}}$ is large. So taking $n_{\text{ev}} = 20,000$ is sufficient.

#### H.3.2 Evaluating the Error of the Test (4)

We set the parameters to be $\delta = 0.1$ and $\pi = \frac{1}{2}\delta$ in our experiments. As explained Appendix G.3, here we no longer take $X^{\text{ev}} = X^{\text{tr}}$. Empirically, taking $X^{\text{ev}} = X^{\text{tr}}$ yields a very bad threshold $\gamma(X^{\text{ev}}, Y^{\text{ev}}, \pi)$.[4] Instead, $X^{\text{ev}}$ is sampled from untouched datapoints other than $X^{\text{tr}}$, and the same applies for $Y$. We still take $n_{\text{ev}} = n_{\text{tr}}$ here, resulting in a total size of $n_{\text{ev}} + n_{\text{tr}} = 2n_{\text{tr}}$. Specifically, when $n_{\text{ev}} \geq 10,000$, computing a $n_{\text{ev}} \times n_{\text{ev}}$ Gram matrix becomes computationally expensive, so we adopt Monte Carlo method to compute $\gamma(X^{\text{ev}}, Y^{\text{ev}}, \pi)$, in which we subsample 10,000 points from $X^{\text{ev}}$ and $Y^{\text{ev}}$ to calculate $\gamma$ and repeat this process 100 times.