# OpenReview forum: "Kernel-Based Tests for Likelihood-Free Hypothesis Testing"
_NeurIPS.cc/2023/Conference — NeurIPS 2023 poster_

### Official Review · Reviewer_x91N · 2023-06-08

**Soundness:** 3 good
**Presentation:** 4 excellent
**Contribution:** 3 good
**Rating:** 6
**Confidence:** 3

**Summary:**

This paper studies a problem in the area of likelihood-free testing, where one is given black-box access to samples $X\sim P_X$, $Y \sim P_Y$ of size n, as well as "real-world data" $Z \sim P_Z$ of size m. The authors propose a new model ("Mixed likelihood-free testing") where it is known that $P_Z = (1 - \nu) P_X + \nu P_Y$ for rate parameter $\nu$, and the goal is to determine whether $\nu = 0$ or $\nu \ge \delta$. They propose a kernel based test statistic for this problem, and analyze its performance theoretically, and also provide lower bounds. They also perform experiments based on Higgs Boson detection and detecting images produced by a diffusion model to support their theoretical claims.

**Strengths:**

This paper is well written, and the problem is well motivated from the point of view of practical applications (Higgs Boson detection, diffusion image detection). The paper has an extensive set of experiments on real data, and there is an interesting theory to support empirical findings. Overall, I think this is an interesting paper.

**Weaknesses:**

The upper and lower bounds in the theory are a little difficult to interpret, especially the dependence on the eigenvalues of the kernel - it would be useful if the authors could provide some examples with simple kernels to illustrate better what the bounds are saying.

Some of the things done in the experiments, such as training a kernel on the test data (especially overlap between training and evaluation data), are not captured/explained by the theory. Also, while Figure 1 seems like it agrees with the theory, it includes all the samples, including ones used for training, calibration, and evaluation. On the other hand, the plot involving only the evaluation samples (Figure 8 in appendix) seems like it agrees more with the naive expectation of $\min\{n, m\} \gtrsim n_{TS}(\alpha, \epsilon, \mathcal P)$. Since the theory is supposed to reflect performance relative to evaluation samples for a fixed kernel, it would be useful if the authors could provide a more detailed explanation of both these plots. It would also be useful if the authors could provide a simple synthetic experiment to illustrate the theoretical bounds for the setting that it is supposed to explain (with a fixed kernel)

The dependence on mis-specification parameter $R$ is not captured by the lower bounds, and Proposition 4.1 (which is about the calibration of the test statistic) assumes that $R=0$. I would be curious to know if proposition 4.1 can be made stronger. In particular, can anything be said about the empirical settings where presumably $R > 0$?

One very interesting thing the authors do in experiments is to use a fraction of the samples from $P_X$ and $P_Y$ to train a kernel that maximizes the (normalized) expectation gap of the statistic. While this seems to work well, there doesn't seem to be a theoretical explanation for why this is a good idea. Of course, while it may not currently be possible to make theoretical claims about a neural net based kernel, is it possible to make a theoretical claim about some simpler model? Or make some abstract claim given black box access to an algorithm that can take samples and generate a kernel with some good properties?

**Questions:**

1) What is the relationship between this problem/likelihood-free testing more generally, and the distribution testing literature in theoretical CS? Can you include a brief comparison?

2) While Proposition 4.1 says that the p-value estimate is good as n_cal-> infty, what is the finite-sample relationship between n_cal, n_eval and the test statistic? Is there something quantitative that can be said here? What happens if $R > 0$?

3) Is it possible to say anything about the kernel training phase theoretically?

**Limitations:**

Yes

---

> ### Author Rebuttal · Authors · 2023-08-10
>
> We greatly appreciate the helpful comments in the review and will make sure to address them in the revision. In the following we will address some of the questions mentioned in the review.
>
> 1. $\textbf{“The upper and lower bounds in the theory are difficult to interpret.”} $
>
> In terms of simple examples, we would like to highlight those given in Section 3.5 as well as the three examples (discrete distributions, Holder-smooth distributions, and Sobolev-smooth densities) of specific kernels provided in Appendix B, which show that our upper bounds recover minimax optimal results in past literature ([1]). Among them, the discrete case could be the simplest example to provide the reader with some intuition. In the revision, a more focused discussion will be included to help readers understand our results by adding explicit evaluations of kernel spectra. Please also see general response #1 for more.
>
> 2. $\textbf{“What is the relationship between LFHT and the distribution testing literature in TCS ?”} $
>
> The closest work to LFHT in the TCS literature is two-sample testing with unequal sample size which is in fact equivalent to LFHT in the regime m > n for discrete distributions (this can be found in [1]). There has also been some work on LFHT in information theory [5]. We recommend the papers [1,3] as a good resource for connections to prior results. To the best of our knowledge, the mLFHT problem hasn’t been considered by the TCS community and we are not aware of related/similar work.
>
> 3. $\textbf{“What is the finite-sample relationship between $n_\mathsf{cal}$, $n_\mathsf{eval}$ and the test statistic? Is there a version of Proposition 4.1 when $R>0$? ”} $
>
> That is an important question that we hadn’t considered before. A naive answer would be to force all the k batches to be disjoint, namely take $k=\lfloor n_\mathsf{cal}/m \rfloor$. In this case, the variance would be of order $\mathcal O(m/n_\mathsf{cal})$ and thus $n_\mathsf{cal} = \omega(m)$ is sufficient for the variance to decay to zero. Improving this naive bound, or alternatively studying the hardness of estimating the p-value could be an interesting direction.
>
> When $R>0$ estimation of the p-value becomes subtle. By definition the p-value of the composite hypothesis is $p:=\sup_{\nu(P)=0}(P(\hat{T}<T))$. However, here the maximizer might not be $P=P_X$, which invalidates the current algorithm. Due to the generality of our setup, designing an algorithm to consistently estimate the p-value in the misspecified setting seems like a challenging open problem.
>
> 4. $\textbf{“Is it possible to say anything about the kernel training phase theoretically?”}$
>
> In the context of two-sample testing, Theorem 6 in [2] shows under mild assumptions, that estimating the kernel by optimizing the objective J is a consistent procedure (i.e. it identifies a maximizing kernel). The same paper (page 3, left column, bottom) includes a justification of using J as the objective that relies on the asymptotic power of the resulting test. In a similar vein we include a heuristic justification of our use of the objective J in Appendix F1. However, these results leave much to be desired. Studying the problem of kernel selection in a simplified/toy setting (even finding the right way to pose the question) might uncover interesting phenomena, although we have not spent time on this.
>
> 6. $\textbf{Figures 1,8 and the trade-off}$
>
> Figure 1 agrees with our theory insofar as it exhibits a clear asymmetric trade-off between n and m with m leveling off at a lower value than n (although we do not claim that our theoretical results strictly imply that this should be the case for trained kernels). We view Fig. 1 as important empirical evidence and motivation for further study of this simulation-experimentation tradeoff in (m)LFHT. We agree that on Figure 8 the precise nature of the trade-off is visually less clear, this may be due to the high numerical precision and/or samples required to observe the trade-off. We are happy to include a synthetic experiment to verify the soundness of our theoretical trade-off in the revision.
>
> $\textbf{References}$
>
> [1] Gerber, P. R., and Polyanskiy, Y. Likelihood-free hypothesis testing. 2022.
>
> [2] Liu, F., Xu, W., Lu, J., Zhang, G., Gretton, A., and Sutherland, D. J. Learning deep kernels for non-parametric two-sample tests. 2020.
>
> [3] Gerber, P.R., Han, Y. and Polyanskiy, Y. Minimax optimal testing by classification. 2023.
>
> [4] Bhattacharya, B. and Valiant, G. Testing closeness with unequal-sized samples. 2015.
>
> [5] Huang, D. and Meyn, S. Classification with high-dimensional sparse samples. 2012.

---

> > ### Comment · Reviewer_x91N · 2023-08-16
> >
> > Thank you for your comments. I will keep my rating as is.

---

### Official Review · Reviewer_m3Zx · 2023-07-03

**Soundness:** 3 good
**Presentation:** 2 fair
**Contribution:** 2 fair
**Rating:** 5
**Confidence:** 2

**Summary:**

This paper introduces a new framework for likelihood-free hypothesis testing (LFHT), denotes as Mixed-LFHT or mLFHT. The setting of mLFHT is as follows: assume $n$ i.i.d. samples from a background disttribution $P_X$, and the same number of samples from a signal distribution $P_Y$. Also, assume we are given another $m$ i.i.d. samples from a mixed distribution $P_Z=\nu P_Y + \left(1-\nu\right)P_X$, and the aim is to "test" and see if $\nu$ is greater than some positive threshold $\delta$. Paper proposes a new kernel-based test statistic for this problem and then analyzes it from both theoretical and experimental perspectives. The main idea is to somehow "map" the empirical distributions of observed data into an RKHS (for some kernel of choice) and then measure the discrepancies of the observed data from both $P_X$ and $P_Y$.

From a theoretical POV, authors claim to acheieve non-trivial sample complexity bounds for the non-asymptotic regime. Both upper and lower-bounds on the error have been established in Thms 3.2 and 3.3, respectively. From an experimental side, authors propose a new algorithm to learn the kernel from data, and also showcase the applicability of their test statistic and algorithm on a number of real-world tasks.

I am not an expert in this particular field and therefore not much aware of other similar works. However, paper is not well-written and seems a little rushed. Regarding the technical contribution, I have a number of questions (please see questions section) before reaching a final conclusion. For now, my vote is borderline reject, with a relatively low confidence.

**Strengths:**

- Paper investigates an important problem for the ML community in general.
- The mathematical tools (specially those from functional analysis and RKHS-related areas) used for the theoretical parts are sufficiently sophisticated and might be interesting for many researchers in this field.
- Paper attempts to propose a new setting which comes with both a) theoretical guarantees and b) applicability on real-world data,
- I did not completely checked the proofs, however, so far I have not found any miscalculations or mathematical flaws.

**Weaknesses:**

- IMO paper is not well-written, and this might be the biggest weakness of this work. Introduction seems short and not that much informative. Also, transitions from one part or section to another do not seem to be smooth enough. There are some cases where an equation or formula is referred to, but in fact it appears (for the first time) at least two pages later which makes the paper a little confusing for a general reader. Positioning of the work w.r.t prior papers is lacking in some ways, and some assumptions are not sufficiently justified. Overall, it seems that authors have tried to compress too much information into a 9-page limited manuscript which has degraded its readability. I suggest a major reconsideration in the writing, or even maybe considering this work for a journal paper instead of a conference.

- One of the main theoretical contributions, as far as I have understood, is the claimed "non-trivial" behavior of the sample complexity bound which is outlined in Theorem 3.2, where authors show a non-symmetic behaviour between $m$ and $n$. However, in L.79 authors mention a similar relation in an existing work (Ref. [18]), which I assume is achieved for a slightly different problem. If this non-symmetric relation has been already discovered, it might affect the contribution level of this paper (please correct me if I am wrong).

- In Thm. 3.3 authors claim to achieve an error lower-bound for mLFHT which matches (more or less) the error upper-bound in Thm 3.2. However, my understanding is that the bound in Thm 3.3 is mainly for the particular kernel-based solution of this paper, and not for mLFHT in general. The actual error lower-bound might be a lot smaller (or not). Again, please correct me if I am wrong.

- I have some questions regarding a number of assumptions and derivation in the paper to see how they might limit the applicability of this solution (please see Questions section).

-----------------------------------------------------------------

Minor comments:

- Caption of Fig. 1: Eq. (4) appears on page 4, but figures is on page 2.

- L.82 to L.89: explanations are vague, please consider some improvements.

- Section 1.2 is a key part of the technical contribution (introducing mLFHT), but there are no prior announcements or motivations for it. Please highlight it more.

- L.105 to L.109: Explanations seem to be non-informative, and can be removed.

- Transition from Section 1.2 to 2.1 is not smooth at all. Also, the math that comes afterwards is unncessarily intense and some of it can be moved to appendix (Specially the "Removing the Bias" part, I guess it won't hurt the paper it this part is moved).

- Again, section 2.2 needs more prior advertisement and explanations.

- L.140 to L.151: explanations seem vague.

- L.154: please consider citing some ref. paper for the theorem, otherwise it might look like it is your contribution.

- L.179: Authors claim that this paper imposes less restrictions on the class of distributions for $P_X$ and $P_Y$ compared to other works. I am not completely sure about this claim (see questions), however, it might be a good idea to advertise it more. For example, authors can bring it up in the Introduction section.

**Questions:**

- Please position your work w.r.t Ref. [18], since they also have achieved a similar non-symmetric and non-aasymptotic relation between $m$ and $n$.

- Please justify the assumptions (i) and (iii) in L.165 and L.167, respectively. Assumption (ii) makes sense, but I assume (i) and (iii) might be limiting. Are they necessary in general (are they fundamentally required)? If yes, then please elaborate. If not, please explain how this will affect the applicability of this method in real-workd tasks.

- In Theorem 3.2, how can we deal with $R$? While reading Section 3.5, I noticed $R$ automatically becomes zero in other existing works since they assume $P_Z=P_X$ or $P_Z=P_Y$. This makes the comparison a little troublesome. Again, is the appearance of $R$ inevitable in your equations? or is it a limitation of this work?

- Related to previous question: In L.101 it has been said that $P_Z\triangleq (1-\nu)P_X+\nu P_Y$. Then, in L.163 we have $\nu=\arg \min_{\nu'} \mathrm{MMD}(P_Z,(1-\nu')P_X+\nu' P_Y)$. Does this mean $R$ is always zero, since $P_Z$ is already assumed to be a convex combination of $P_X$ and $P_Y$? This part is confusing, please clarify.

**Limitations:**

-

---

> ### Author Rebuttal · Authors · 2023-08-08
>
> We appreciate the reviewer’s comments on our writing as well as the ‘minor comments’; our revision will incorporate all of the helpful suggestions. The page limit is indeed challenging, but we believe we are able to make the improvements necessary for good readability. Below, we address some more specific questions brought up in the review.
>
> 1. $\textbf{ “Comparisons to Ref. [18].”}$
>
> In short, we study a generalization of the problem studied in [18], and we obtain results that imply many of the results therein. Indeed, [18] was the first to identify the trade-off between m and n, but we show that this phenomenon apply much more generally and are the first to empirically demonstrate it in real datasets (beyond information theoretically worst case constructions) via a novel, theoretically provable, efficient algorithm. Please see also our general response #2.
>
> 2. $\textbf{ “Thm 3.3 applies only to the particular kernel-based solution and not mLFHT in general.”}$
>
>
> Our lower bounds in Thm 3.3 apply to the mLFHT problem in general (not algorithm based). In other words, any algorithm for mLFHT must use a number of samples greater than the bound in Thm 3.3. Our proof (Appendix E) uses information-theoretic techniques and bounds the total variation distance between classes of constructed hard instances involving the eigenfunctions of the kernel. We will clarify this in the revision.
>
> 3. $\textbf{ “Justify the assumptions (i) and (iii) in L.165 and L.167.”}$
>
>
> Assumption (i) is crucially required for our results and we believe is somewhat mild (the upper bound on the density). Note also that the base measure can be selected arbitrarily, giving a lot of flexibility in what is considered “bounded” density (e.g. the base measure could become singular at the border of the manifold, etc). As for (iii), it is less restrictive than requiring that R=0 (i.e. that P_Z is exactly a mixture of P_X and P_Y) which was done in prior work [18]. Therefore, (iii) is a relaxation of a commonly used assumption. Specifically, tuning $R$ controls how misspecified the problem is allowed to be. See also our general response #3 about (iii).
>
> 4. $\textbf{ Questions 3-4}$
>
> In L101-105 we describe a not fully general version of mLFHT (to keep the introduction simple), which corresponds to the case R=0 of L162-167. Our referencing equation (mLFHT) in Theorem 3.2 indeed makes this confusing, we will correct this. The introduction of $R$ should be viewed as an extension (or relaxation, cf. previous part) not as a limitation, we think. Please also see general response #3 for more.

---

> > ### Comment · Reviewer_m3Zx · 2023-08-16
> >
> > Thanks for the detailed response.
> > My concerns w.r.t. lower-bound are alleviated. However, my other two concerns are still remaining:
> > 1) paper (at least in parts) should be rewritten to reflect the fact that the proved non-symmetric relation between $m$ and $n$ is not entirely new, and the current paper has only shown its applicability in a more general setting.
> > 2) The more general setting in this work has introduced at least one new parameter, denoted as $R$, which appears in almost all the bounds and is in fact a fundamental property of the problem set. However, authors have not discussed this parameter in details. Is it necessarily needed? how tight are the bounds w.r.t. $R$?
> >
> > Overall, assuming the authors will correct all the shortcomings mentioned during the reviews (specially the confusing parts between L.101-105 and L.163), I raise my score to borderline accept.

---

> > > ### Author Response · Authors · 2023-08-17
> > >
> > > Thank you for your effort in reviewing our paper. We will make sure to edit the paper according to your advice. Let us address your two points below.
> > >
> > > 1) We hope we clearly identified that our work is an extension of the prior art, e.g. quote from the second paragraph of the paper: ''Recently, Gerber and Polyanskiy (2022) proposed likelihood-free hypothesis testing (LFHT) as a simplified model and found minimax optimal tests for a range of non-parametric distribution classes, thereby identifying a fundamental simulation-experimentation trade-off between the number of simulated observations n and the size of the experimental data sample m. Here we extend Gerber and Polyanskiy (2022) to a new setting designed to model experimental setups more truthfully and derive sample complexity (upper and lower bounds) for kernel-based tests over non-parametric classes.'' However, we will do more and promote this fact in the abstract. Our general response #2 contains more details related to this question.
> > >
> > >
> > > 2) Indeed, in this work we introduced at least two more parameters: the $\delta$ in (mLFHT), which in our opinion is crucial, and the robustness parameter $R$. We acknowledge that we have not identified optimal tradeoffs for either of them: it is to be a subject of future work. Furthermore, in all honesty, we did not view $R$ as very important (unlike $\delta$), since it was introduced only with the goal of demonstrating that our results are at least somewhat robust to misspecification; it is not
> > > $\textit{needed}$ in the sense that our work mostly, if not only, concerns the case $R=0$. We thank you for making us think deeper about this parameter and hopefully, we’ll get optimal results in the future. Our general response #3 contains more details related to this question.

---

### Official Review · Reviewer_92Pk · 2023-07-21

**Soundness:** 3 good
**Presentation:** 3 good
**Contribution:** 2 fair
**Rating:** 5
**Confidence:** 2

**Summary:**

The paper addresses a likelihood-free hypothesis testing (LFHT) scheme, where the data distribution is a mixture of two distributions. The null hypothesis is given by one of these distributions, whereas the alternative hypothesis is that the mixture coefficient is lower-bounded by some delta.  The motivation for such a scheme is that it is tied to some applications in particle physics, for example, the Higgs boson discovery. This is a generalization of the original LFHT scheme, which is recovered if delta=1. Goal of the paper is to characterize the tradeoff between n, i.e., the size of the dataset and m, the number of samples available from the mixture data distribution available at test time. Specifically, the authors derive lower and upper bounds on the minimal sample complexity. They also empirically verify this trade-off for kernel based hypothesis testing, where the kernels are parameterized via neural networks.

**Strengths:**

An interesting result is the dependency between n and m for which the authors derive lower and upper bounds, with a non-trivial trade-off  between these quantities. It seems though that this was already addressed in prior work, but this is not my research area and I may be missing other contributions.

**Weaknesses:**

It seems that the applications for the proposed mixture LFHT are quite limited. In particular, the Higgs boson discovery is a very specific application, and therefore it is unclear whether this work extends to other key applications.

The authors mention in Section 4 that the results from Theorem 3.2 cannot be directly applied because of the dependency of data and kernel. This is not clear to me. Isn’t the main idea of a kernel that it is learned from the data? It also seems to be the reason why their proposed generalization bounds cannot be included in Fig. 1 as a reference, which questions the usefulness of these bounds as fundamental limits of the scenario at hand.

The statement in line 274 seems like a trivial observation, typically the validation set is always chosen much smaller than the training set.

**Questions:**

See weaknesses.

**Limitations:**

Does not apply.

---

> ### Author Rebuttal · Authors · 2023-08-08
>
> We greatly appreciate the helpful comments in the review and will make sure to address them in the revision. In the following we will address some of the questions mentioned in the review.
>
> 1.$\textbf{”It seems that the applications for the proposed mixture LFHT are quite limited.”}$
>
> Our motivation for defining mLFHT is in fact very much practically driven: if one tries to model high-energy physics (LHC) or gravitational waves experiments (LIGO) those typically do have an alternative (non-null) hypothesis under which samples come from a mixture of noise and signal distributions. One good illustration of the interest of the physics community is [4] with over 8000 citations which studies a problem essentially equivalent to mLFHT in an asymptotic and parametric setting. Furthermore, since it is both an extension of LFHT and two-sample testing, mLFHT shares many applications arising therein (e.g. generative model comparisons, bioinformatics, etc, [1, 2]). The main objective of this work is to propose a theoretically provable efficient algorithm that is also practical in handling real-world data (which is well-modeled by our mixture framework), including the two examples shown in this paper. Specifically, what is crucial in the mLFHT setting is that despite the fact that the weight of the “signal” component in the alternative hypothesis can be very small, the number of simulation samples does not need to be large because the simulation samples can be forced to come directly from the signal component of the mixture. This prevents a certain kind of “curse of dimensionality” as we attempted to explain in the last paragraph of Section 3. We will make sure to explain the motivation and novelty of mLFHT more clearly in a revision.
>
> 2. $\textbf{“Connection of Figure 1 and Theorem 3.2.”}$
>
> As you point out, our theoretical upper bounds rely on an a priori fixed kernel, while in our empirical experiments we train the kernel based on part of the data. Instead of using artificial experiments with a fixed kernel to empirically illustrate the soundness of our sample complexity bounds, we chose to use a state-of-the-art algorithm. The purpose of Fig. 1 is to show that an asymmetric simulation-experimentation tradeoff arises on real data and with complex real-world (kernel-training) algorithms beyond the theoretical minimax fixed kernel setting. We view Fig. 1 as important empirical evidence and motivation for further study of this simulation-experimentation tradeoff in (m)LFHT.
>
> 3. $\textbf{“L274 is trivial”}$
>
> We believe that the observation is not entirely trivial since $X^{ev}, Y^{ev}$ are not used for validation in the traditional sense but rather serve as samples in estimating (evaluating) the test statistics conditioned on the kernel trained on a held-out set.
>
> $\textbf{References}$
>
> [1] Lintusaari, J., Gutmann, M. U., Dutta, R., Kaski, S., and Corander, J. Fundamentals and recent developments in approximate Bayesian computation. 2017.
>
> [2] Bounliphone, W., Belilovsky, E., Blaschko, M. B., Antonoglou, I., and Gretton, A. A test of relative similarity for model selection in generative models. 2016
>
> [3] Gerber, P. R. and Polyanskiy, Y. Likelihood-free hypothesis testing. 2022.
>
> [4] Cowan, G., Cranmer, K., Gross, E. and Vitells, O. Asymptotic formulae for likelihood-based tests of new physics. 2011.

---

### Official Review · Reviewer_5XYH · 2023-07-22

**Soundness:** 3 good
**Presentation:** 3 good
**Contribution:** 3 good
**Rating:** 7
**Confidence:** 3

**Summary:**

This paper proposes a test statistic that can be used for likelihood-free hypothesis testing [LFHT] (for distributions $P_X, P_Y$, given a sample from $P_Z$, decide if $P_Z = P_X$ versus $P_Z = P_Y$) and mixed likelihood-free hypothesis testing [MLFHT] (for $P_Z = (1 - \nu) P_X + \nu P_Y$, decide if $ \nu = 0 $ versus $ \nu \ge \delta$).

The test statistic is defined as the MMD distance between $P_Y$ and $P_X$ for an appropriately chosen kernel $K$. The paper then shows an upper and lower bound for these two testing problems. These depend on a base measure $\mu$ and a kernel $K$, both of which can be chosen, such that if $e_j$ are eigenfunctions of $\mu$, then $K(x, y) = \sum_j \lambda_j e_j(x) e_j(y) $. The bounds are then stated in terms of $\lambda$ (either $\mid\mid \lambda \mid \mid_\infty $ or $ \mid \mid \lambda \mid\mid_2$).



**Strengths:**

- The results are very general with reasonable assumptions. The authors assume that the distributions $P_x, P_y, P_z$ have bounded densities wrt the base measure $\mu$, and the sample complexity scales linearly with this bound on the density. This is in contrast to existing work, which assumes more descriptive assumptions on these distributions.

- There is empirical support for the asymmetric sample complexity between the train sample $n$ and the test sample $m$.

- The lower bounds are general, in that although they are minimax, they apply for any choice of $\mu, K$.

- The upper bound allows for distribution mismatch, i.e., the test sample need not be a perfect mixture of $P_X, P_Y$. The bound holds as long as the true distribution is close in MMD to some mixture of $P_x, P_y$.

- This approach also allows for learning a kernel from the data, although the theory bounds don't hold in this case as it violates independence between choice of the kernel and the training samples.



**Weaknesses:**

- The bounds are a little non-intuitive / roundabout, as they depend only on the base measure $\mu$ and the kernel $K$, both of which do not directly depend on the hypothesis class containing $P_x, P_y$. From my understanding, the only place the hypothesis class shows up is that it must have bounded density with respect to the measure $\mu$.

- The novelty of the results over existing work is not explained fully. Is the main contribution of the work bounds for the mixed-distribution testing? Or is the test statistic and the analysis also entirely new?

- As mentioned by the authors, there is gap between the upper and lower bounds for the training set size $n$ and its dependence on the mixture coefficient $\delta$. However, I think this is acceptable and a good open question.

**Questions:**

See weaknesses section.

**Limitations:**

The authors discuss differences between the upper and lower bounds, but apart from that, there's no discussion of limitations. There should be more discussion about drawbacks of this approach wrt existing approaches -- surely there must be some based on how general these results are in comparison to existing work.

---

> ### Author Rebuttal · Authors · 2023-08-08
>
> We greatly appreciate the helpful comments in the review and will make sure to address them in the revision. In the following we will address some of the questions mentioned in the review.
>
> 1. $\textbf{“The bounds are a little non-intuitive / roundabout.”}$
>
> Please see our general response #1. We believed that having a mild assumption on the hypothesis class helps our setting to model real-world tasks better.
>
> On the other end, indeed the dependency between the trained kernel and finite observations from the hypotheses distributions lacks a satisfying theory. We consider this as a major open problem worth investigating in not only (m)LFHT but also other applications of kernel methods (e.g. two-sample testing [3]).
>
> 2. $\textbf{“The novelty of the results over existing work is not explained fully.”}$
>
> Please see our general response #2 on our novelty over [1]. To specifically address the other points, a version of our MMD test statistic was indeed proposed in prior work (e.g., [2]). However, the theoretical analysis of the MMD statistic (especially non-asymptotically) on the (m)LFHT problem is novel to our knowledge. We will make sure the novelties are highlighted and contrasted properly in the revision.
>
> 3. $\textbf{“Discussion of limitations.”}$
>
> Beyond limitations in Appendix L, some other limitations include our kernel optimization lacking sufficient theory (and therefore, the upper bound must rely on fixed kernels), dependency on certain parameters being open, and several open questions when model misspecification ($R>0$) is present. We will make sure to include a more thorough discussion in the revision.
>
> $\textbf{References}$
>
> [1] Gerber, P. R. and Polyanskiy, Y. Likelihood-free hypothesis testing. 2022.
>
> [2] Bounliphone, W., Belilovsky, E., Blaschko, M. B., Antonoglou, I., and Gretton, A. A test of relative similarity for model selection in generative models. 2015.
>
> [3] Liu, F., Xu, W., Lu, J., Zhang, G., Gretton, A., and Sutherland, D. J. Learning deep kernels for non-parametric two-sample tests. 2020.

---

> > ### Comment · Reviewer_5XYH · 2023-08-18
> >
> > Thank you for the rebuttal. I have read the other reviews and responses. I agree with the reviewers regarding clarity of the results, and I think including the suggested edits will help improve the paper. While there do remain open questions related to this work, I think the contributions warrant an accept, and I'll keep my score as is.

---

### Author Rebuttal · Authors · 2023-08-08

We greatly appreciate the valuable suggestions given by reviewers and will revise our manuscript accordingly. In this response, we would like to address several comments that appear in multiple reviews.

1. $\textbf{“Interpretation of the bounds and its dependence on kernel, base measure, hypotheses”}$

Due to the minimal assumptions for the generality of our results, our bounds can indeed be challenging to interpret. We note that the dependence on the kernel spectrum is unavoidable as can be seen from comparing our upper and lower bounds. To build intuition, in Section 3.5 and Appendix B our upper bound (Thm 3.2) is applied with some explicit kernels (Gaussian kernels/discrete kernels) and to classical nonparametric classes such as smooth densities and bounded discrete PMFs. These applications recover minimax optimal sample complexity shown in past work [1,2]. We hope these concrete examples help in interpreting our results within a broader context. We will take more care in the revision to emphasize these.

2. $\textbf{“Novelty over [1] (Ref. [18] in our paper)”} $

[1] is an important inspiration for our study, however, we innovate over [1] in several ways, which we try to summarize below.

i) $\textbf{New problem setting.}\quad$ [1] derived the minimax tradeoff for several specific classes of distributions that are different from our own (they studied smooth densities on finite-dim spaces and discrete PMFs under TV distance, whereas we study $\underline{\text{unconstrained densities}}$ but under kernel distance). Moreover, our work extends [1] to a more general setting (mLFHT), which on one hand is more practical, and on the other hand, exhibits a different tradeoff (e.g. mitigates the curse of dimensionality in # of simulated samples (L236)). This setting arises in several scientific disciplines (HEP, LIGO, etc) and is more realistic (without assuming smoothness).

ii) $\textbf{More general results.}\quad$ Although under different settings, our upper bound results, applied with specific kernels (Appendix B), recover several of [1]’s key (minimax optimal) results for LFHT as well as several of [2] in two-sample testing. Thus, our theoretical bounds are more general while still being optimal when specialized to prior settings.

iii) $\textbf{Practical algorithm and empirical trade-off.}\quad$ Our (novel) kernel-based algorithm is efficient and able to solve real-world tasks, whereas the tests considered in [1] require discretizing space into a large (exponential in dimension) number of bins and are thus impractical (though minimax optimal). Furthermore, our experiments confirm the validity of the theoretical insight that one can trade off m versus n and that in general one needs more simulation samples n than real samples (asymmetry of the trade-off).

We will make sure that our contributions are clarified and emphasized in the revision.

3. $\textbf{“Dependence on R” and “Assumption (iii)”}$

First, we want to clarify that L.101 is informal and, in fact, the special case ($R=0$) of the mLFHT problem stated in Section 3.2. The latter can be formally written as
\begin{align*}
	H_0: P_Z\in \lbrace P_Z : \nu(P_Z)=0 \rbrace \quad \text{versus} \quad H_1: P_Z\in\lbrace P_Z:\nu(P_Z)\geq\delta\rbrace
\end{align*}
where $\nu(P_Z)=\arg\min_{\nu’}\mathrm{MMD}(P_Z,(1-\nu’)P_X+\nu’ P_Y))$. Intuitively, (iii) says that “the test distribution P_Z should not be too far away from some mixture of labeled data,” where $R$ governs how mis-specified we allow the problem to be. Thus, (iii) shouldn't be thought of as an 'assumption', but rather as a relaxation of the assumption (present in prior work on LFHT) that P_Z is an exact mixture of P_X and P_Y.

We want to clarify that the major focus of our paper is on the case when $R=0$, i.e. when there is no model mis-specification and $P_Z$ comes from a mixture of $P_X$ and $P_Y$, as is assumed in many applications and past literature. Our theory and experiments cover this scenario perfectly by simply plugging in $R=0$ in Thm 3.2.

In Thm 3.2, we included the more general case where $R$ can be positive in order to present the most general result possible. The dependence on $R$ is not a crucial part in the theory or experiments. While a direct controlled study involving $R$ would be hard (since it depends on the kernel learned from samples), we believe that further investigation of (m)LFHT involving non-trivial model mis-specification is an interesting open direction worth investigating.

$\textbf{References}$

[1] Gerber, P. R. and Polyanskiy, Y. Likelihood-free hypothesis testing. CoRR, abs/2211.01126, 2022.

[2] Li, T. and Yuan, M. On the optimality of gaussian kernel based nonparametric tests against smooth alternatives. arXiv preprint arXiv:1909.03302, 2019

---

### Decision · Program_Chairs · 2023-09-21

**Decision:**

Accept (poster)

**Comment:**

This paper studies the likelihood-free testing problem, recently introduced by Gerber and Polyanskiy, motivated by practical questions in physics experiments. The contributions were well appreciated by the reviewers, and the authors used the rebuttal phase successfully to clarify misunderstandings about their theoretical results.